# TATA and paused promoters active in differentiated tissues have distinct expression characteristics

Vivekanandan Ramalingam[1,2,†,‡] (iD), Malini Natarajan[1,†,§] (iD), Jeff Johnston[1,¶] (iD) & Julia Zeitlinger[1,2,*] (iD)

## Abstract

Core promoter types differ in the extent to which RNA polymerase II (Pol II) pauses after initiation, but how this affects their tissue-specific gene expression characteristics is not well understood. While promoters with Pol II pausing elements are active throughout development, TATA promoters are highly active in differentiated tissues. We therefore used a genomics approach on late-stage *Drosophila* embryos to analyze the properties of promoter types. Using tissue-specific Pol II ChIP-seq, we found that paused promoters have high levels of paused Pol II throughout the embryo, even in tissues where the gene is not expressed, while TATA promoters only show Pol II occupancy when the gene is active. The promoter types are associated with different chromatin accessibility in ATAC-seq data and have different expression characteristics in single-cell RNA-seq data. The two promoter types may therefore be optimized for different properties: paused promoters show more consistent expression when active, while TATA promoters have lower background expression when inactive. We propose that tissue-specific genes have evolved to use two different strategies for their differential expression across tissues.

**Keywords** effector genes; gene expression noise; Pol II pausing; scRNA-seq; TATA promoter

**Subject Categories** Chromatin, Transcription & Genomics; Development

**Mol Syst Biol. (2021) 17: e9866**

## Introduction

The core promoter is the ~ 100 bp sequence surrounding a gene's transcription start site (TSS) that facilitates the assembly of the transcription machinery and Pol II transcription (Smale & Kadonaga, 2003; Haberle & Stark, 2018). Pol II transcription may be stimulated in a tissue-specific manner by activation signals from enhancer sequences (Banerji *et al*, 1981; Spitz & Furlong, 2012), but a core promoter may also produce basal or background levels of transcription in the absence of an activation signal (Kim *et al*, 1994; Verrijzer & Tjian, 1996; Juven-Gershon *et al*, 2006). Ideally, a promoter produces only minimal background expression in the inactive state, is highly responsive to enhancers, and reliably produces the desired level of transcription in the active state.

A core promoter element that strongly promotes Pol II initiation is the TATA box (Patikoglou *et al*, 1999; Reeve, 2003), an ancient core promoter element present in archaea, fungi, plants, and animals (Patikoglou *et al*, 1999; Reeve, 2003). TATA is bound by TATA-binding protein (TBP) and helps assemble the pre-initiation complex (Nikolov *et al*, 1992; Kim *et al*, 1993; Patikoglou *et al*, 1999). After initiating transcription, Pol II may then pause 30–50 bp downstream of the TSS, before being released into productive elongation (Adelman & Lis, 2012).

Based on work in *Drosophila*, core promoter elements not only influence Pol II initiation, but also Pol II pausing. Promoters with very stably paused Pol II are enriched for downstream pausing elements such as the Pause Button (PB) and the Downstream Promoter Element (DPE) (Burke & Kadonaga, 1997; Lim *et al*, 2004; Hendrix *et al*, 2008; Gaertner *et al*, 2012; Shao & Zeitlinger, 2017). TATA promoters often show minimal Pol II pausing but the evidence is conflicting and predominantly based on cultured cells (Gilchrist *et al*, 2010; Chen *et al*, 2013; Shao & Zeitlinger, 2017; Krebs *et al*, 2017). Swapping core promoter elements or the entire promoter alters the amount and duration of Pol II pausing in *Drosophila* embryos and cultured cells (Lagha *et al*, 2013; Shao *et al*, 2019).

The amount of Pol II pausing at a promoter appears to influence the expression characteristics. Promoters with high occupancy of paused Pol II are prevalent among genes that are highly regulated during development (Muse *et al*, 2007; Zeitlinger *et al*, 2007; Gaertner *et al*, 2012) and mediate more synchronous gene expression between cells (Boettiger & Levine, 2009; Lagha *et al*, 2013). Without well-timed gene activation, coordinated cellular behaviors such as gastrulation may not proceed properly (Lagha *et al*, 2013). TATA promoters on the other hand are often associated with higher expression variability (Raser & O'Shea, 2004; Blake *et al*, 2006; Tirosh *et al*, 2006; Lehner, 2010; Hornung *et al*, 2012; Li & Gilmour, 2013; Sigalova *et al*, 2020). However, the expression characteristics of TATA promoters in developing embryos are less understood.

1   Stowers Institute for Medical Research, Kansas City, MO, USA
2   Department of Pathology and Laboratory Medicine, University of Kansas Medical Center, Kansas City, KS, USA
    *Corresponding author. E-mail: jbz@stowers.org
    †These authors contributed equally to this work
    ‡Present address: Department of Genetics, Stanford University, Stanford, CA, USA
    §Present address: Department of Molecular Biology, Cell Biology and Biochemistry, Brown University, Providence, RI, USA
    ¶Present address: Center for Pediatric Genomic Medicine, Children's Mercy, Kansas City, MO, USA

Across metazoans, genes with TATA elements are particularly enriched among effector genes, the genes responsible for the structure and function of differentiated tissues (Schug *et al*, 2005; Carninci *et al*, 2006; FitzGerald *et al*, 2006; Engström *et al*, 2007; Lenhard *et al*, 2012; FANTOM Consortium *et al*, 2014). Effector genes start to be expressed primarily at later stages of embryogenesis when cells begin differentiation into morphologically distinct tissues (Erwin & Davidson, 2009). These stages are typically not well studied, and thus, whether TATA promoters confer effector genes different expression characteristics is not clear.

Here, we systematically analyzed the relationship between promoter types and gene expression in differentiated tissues of the late *Drosophila* embryo, where both TATA and paused promoters are active. We mapped the gene expression programs of all cell types using single-cell RNA-seq (scRNA-seq) and determined the occupancy of Pol II in a tissue-specific fashion. Our analysis revealed large differences in Pol II pausing between the promoters of effector genes and showed that TATA promoters are strongly enriched among effector genes with minimal Pol II pausing. Notably, scRNA-seq revealed that TATA genes have higher expression variability but lower background expression than paused promoters and that this property correlates with lower chromatin accessibility. We propose that different promoter types are optimal for different expression properties and discuss the mechanisms by which these differences in promoter function occur.

## Results

### Characterization of the tissue-specific expression programs in the late *Drosophila* embryo using single-cell RNA-seq

To obtain an unbiased global view of the gene expression programs in differentiated tissues, we performed scRNA-seq on dissociated cells from late *Drosophila* embryos (Fig 1A). We chose embryos at stage 16 (14–14.5 h after egg deposition) when the tissues are fully formed but the outside cuticle is not yet developed enough to hamper the dissociation of the cells. After processing the cells through a 10× Genomics Chromium instrument (Klein *et al*, 2015; Macosko *et al*, 2015; Zheng *et al*, 2017), we obtained the expression profiles of approximately 3,500 cells. Cells prepared and sequenced from two separate batches yielded results that were highly similar with regard to data quality and results from clustering (Appendix Fig S1).

To identify the tissues to which each scRNA-seq cluster belongs, we correlated the scRNA-seq data with the large-scale *in situ* hybridization data from the Berkeley *Drosophila* Genome Project (BDGP) (Tomancak *et al*, 2002; Tomancak *et al*, 2007; Hammonds *et al*, 2013) (Fig 1A). For ambiguous clusters, we analyzed the occurrence of known tissue markers and manually merged or separated clusters such that they better matched anatomical structures. In this manner, we obtained scRNA-seq data for 16 tissues of the late *Drosophila* embryo: central nervous system (CNS), peripheral nervous system (PNS), glia, germ cells, epidermis, trachea, muscle, dorsal vessel, fat body, plasmocytes, crystal cells, ring gland, salivary gland, gastric cecum, midgut and malpighian tubules (Fig 1B; Dataset EV3). In order to make this classification useful for future studies, we also identified marker genes for each tissue, some of which were previously known (Fig 1B and Appendix Fig S2).

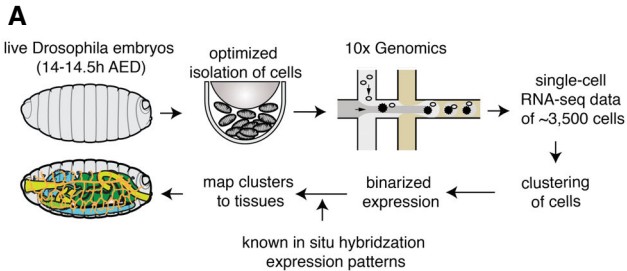

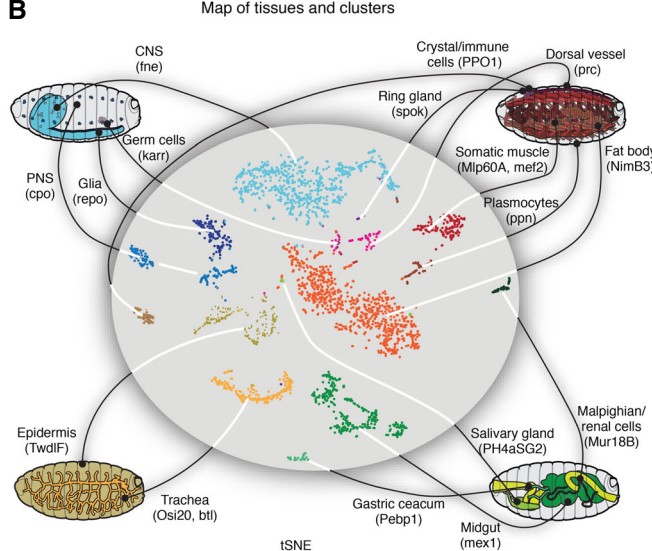

**Figure 1. scRNA-seq captures the expression profiles of effectors genes in the late stages of *Drosophila* embryogenesis.**

A Single cells were isolated from *Drosophila* embryos 14–14.5 h after egg deposition (AED). Isolated cells were processed through a 10× Genomics instrument. After sequencing the resulting libraries, the reads were aligned and processed using the standard pipeline from 10× Genomics. The single-cell gene expression profiles were used to map the cells to known cell types by comparing against the available *in situ* hybridization patterns from the Berkeley *Drosophila* Genome Project.

B A tSNE projection of the scRNA-seq data is shown in the middle, and the known tissues to which the clusters were assigned to are graphically illustrated outside. Marker genes for each tissue type are shown in parentheses.

### Effector genes have different Pol II occupancy patterns across tissues

We next asked what promoter type is used by effector genes in the late *Drosophila* embryo. To distinguish effector genes from housekeeping genes and developmental genes, we defined effector genes by their late upregulation during embryogenesis (> 5×, $P < 0.05$ from 2–4 h to 14–17 h), which yielded 1,527 genes (Datasets EV1 and EV2, Materials and Methods). As control groups, we also defined ubiquitously expressed housekeeping genes (647 genes), as well as developmental genes that are highly paused throughout embryogenesis (772 genes; Dataset EV1) as defined previously (Gaertner *et al*, 2012).

These late-induced genes were enriched for GO terms of tissue-specific biological functions, e.g., synaptic transmission and

chitin-based cuticle development (Fig EV1A), consistent with them being effector genes. They were also under-represented for sequence motifs found in housekeeping genes and enriched for TATA consensus motifs (Fig EV1C). However, sequence motifs typically found in paused developmental promoters such as DPE and PB were also significantly enriched, suggesting that these genes may also be induced by the paused promoter type (Fig EV1C; Dataset EV2).

We therefore set out to characterize these promoters experimentally by performing Pol II ChIP-seq experiments on a variety of tissues isolated from the late *Drosophila* embryo. Using the INTACT method (Deal & Henikoff, 2011; Bonn *et al*, 2012), nuclei from the tissue of interest were genetically tagged for biotin labeling and isolated from fixed embryos (14–17 h) with the help of streptavidin-coupled magnetic beads (Fig 2A). The following six tissues were analyzed: neurons (using *elav*-Gal4), glia (using *repo*-Gal4), muscle (using *mef2*-Gal4), trachea (using *btl*-gal4), and epidermis and gut (using enhancer trap-Gal4 lines 7021 and 110394, respectively, see Materials and Methods; Fig 2B).

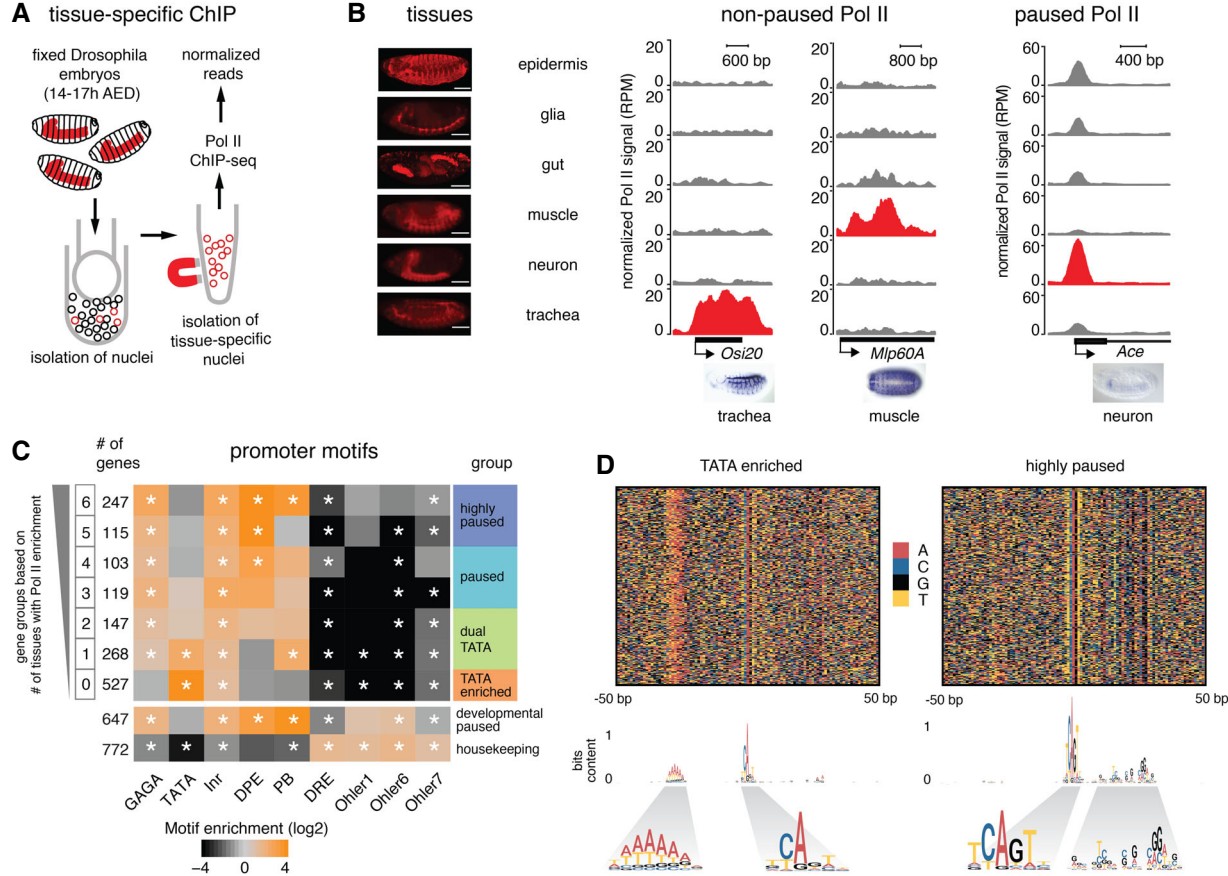

**Figure 2.  Tissue-specific Pol II ChIP-seq shows differences in Pol II occupancy patterns at effector genes.**

A   Tissue-specific ChIP-seq was done by isolating nuclei from specific tissues (shown in red) by expressing the *Escherichia coli* biotin ligase (BirA) and the biotin ligase recognition peptide (BLRP) fused with a nuclear envelope-targeting sequence in the tissue of interest. This allows the isolation of nuclei from the tissue of interest using streptavidin magnetic beads.

B   Pol II ChIP-seq was performed in six different tissues shown in the left panel (scale bar - 100 μm). The middle and the right panels show the read-count normalized Pol II ChIP-seq tracks (RPM) from the six tissues at individual genes. For each gene, gray and red tracks indicate non-expressing tissues and expressing tissue, respectively. The middle panel shows the Pol II profile at two non-paused genes, which have Pol II only in the expressing tissues. The expression is limited to specific tissues as shown in the *in situ* images from BDGP. The right panel shows the Pol II profile at a paused gene, which has Pol II in all observed tissues, although the expression is limited to specific tissues as shown in the *in situ* images from BDGP. Paused Pol II is generally highest in the tissue with the highest expression. We also found systematic differences between samples; thus, some tissues have generally higher enrichments than others, presumably because they are easier to cross-link.

C   Identified effector genes were grouped into seven groups based on Pol II penetrance, i.e. the number of tissues in which Pol II enrichment is above background (calculated in a window starting from the TSS and ending 200 bp downstream). Genes that are highly paused throughout embryogenesis (developmental paused) (Gaertner *et al*, 2012) and housekeeping genes are shown as a control. Core promoter elements are differentially enriched across the groups (Fisher's exact test with multiple-testing correction, *$P < 0.05$), allowing us to classify promoter classes based on Pol II penetrance (highly paused, paused, dual TATA, TATA enriched). The highly paused group is defined by Pol II enrichments in 5–6 tissues and is similar to the developmental paused genes. TATA enrichment is found in the groups with Pol II enrichment in 0 or 1 tissues.

D   Sequence heat map plots show clear and consistent motif differences between the TATA-enriched and highly paused gene groups. The information content of the motifs is plotted as a sequence logo below, revealing a degenerative TATA box, an Inr with or without G, and downstream pausing elements all of which are consistent with previous results (Shao *et al*, 2019).

The Pol II ChIP-seq tracks from the six tissues confirmed that the ChIP-seq data are tissue-specific. For example, the tracheal gene *Osi20* and the muscle gene *Mlp60A* showed high Pol II occupancy in the trachea and muscle samples, respectively, but not in the other tissues (Fig 2B middle panel). Global analyses were also consistent with a high concordance between Pol II occupancy and scRNA-seq (Appendix Fig S3). The Pol II occupancy was however not always tissue-specific. Some genes showed high Pol II occupancy at the promoter in many or all tissues, despite being expressed in a very tissue-restricted fashion. For example, expression of *Ace* is restricted to neuronal populations but showed very high Pol II promoter occupancy in all tissues (Fig 2B right panel). Moreover, the Pol II pattern along the gene was indicative of Pol II pausing since the Pol II occupancy peaks at the pausing position (30–50 bp downstream of the TSS) and is not detected at the gene body (Fig 2B right panel).

These results suggest that there are two types of tissue-specific promoters that are regulated in a fundamentally different fashion. At one type of promoter, Pol II is recruited only in tissues where the gene is expressed and proceeds toward productive elongation without detectable pausing (Fig 2B middle panel). On the other end of the spectrum is a promoter type where Pol II is widely recruited and found paused across all tissues, and Pol II only proceeds toward productive elongation in the tissues where the gene is expressed (Fig 2B right panel).

## Pol II penetrance across tissues separates TATA and paused effector genes

We next asked whether the different Pol II occupancy patterns across tissues could distinguish between promoter types. We classified genes based on their Pol II penetrance (Fig 2C), defined as the number of tissues (from 0 to 6 tissues) in which Pol II is detected around the TSS above background (> 2-fold signal over input; Dataset EV2). Genes with the highest Pol II penetrance (5–6 tissues) were strongly enriched for pausing elements such as GAGA, DPE, or PB (362 highly paused genes), similar to developmental paused genes (Fig 2C; Dataset EV2). On the other hand, genes with the lowest Pol II penetrance (0 tissues) were highly enriched for TATA elements (527 TATA-enriched genes). Gene groups with intermediate Pol II penetrance had weaker enrichments for both types of elements (222 paused genes, 415 dual TATA genes that had both TATA and pausing elements). Although TATA-enriched genes did not have detectable Pol II occupancy, they were nevertheless transcribed according to RNA-seq data (Fig EV1E). This suggests that Pol II may be hard to detect at some TATA genes, presumably because Pol II does not pause and significant levels can only be detected with high levels of transcription.

These results suggest that Pol II penetrance across tissues is another measurement for Pol II pausing that can be used to classify promoter types. Consistent with this, the Pol II penetrance correlates with Pol II pausing when measured by the pausing index (Fig EV1D; Dataset EV2). Furthermore, we confirmed that Pol II penetrance was not biased by expression levels (Fig EV1E; Dataset EV2), although slight differences in the timing of the induction of the TATA-enriched and paused gene groups were detected (Appendix Fig S4; Dataset EV2).

To analyze the promoter groups in more detail, we aligned the promoter groups based on the TSS identified by CAGE data from late *Drosophila* embryos (Hoskins *et al*, 2011). We then visualized the sequence composition using color plots and generated consensus motifs for different promoter groups (Fig 2D; Dataset EV2). This revealed that the majority of promoters in the TATA-enriched group indeed showed TATA-like elements at the expected position of −30 bp upstream of the TSS, as well as a weak Initiator sequence (CA), consistent with previous data (Shao *et al*, 2019). In contrast, the highly paused group showed a strong Initiator sequence (TCAGT) with a G at the +2 position, which has been shown to promote Pol II pausing (Shao *et al*, 2019). In addition, these promoters had a well-positioned G-rich sequence pattern downstream of the TSS around the site of Pol II pausing (Fig 2D). When we performed the same analysis on the previously identified developmental paused genes, the pattern was strikingly similar (Fig EV1F). This indicates that the highly paused promoters during development and in differentiated tissues are functionally equivalent. The functional equivalence is further supported by their similarity in pausing index, gene length, and promoter shape (Fig EV1G and H; Dataset EV2).

Since effector genes have previously been associated with TATA promoters (Schug *et al*, 2005; Carninci *et al*, 2006; FitzGerald *et al*, 2006; Engström *et al*, 2007; Lenhard *et al*, 2012; FANTOM Consortium *et al*, 2014), we asked whether the functions of the highly paused genes also point to them being effector genes. GO analysis revealed functional categories such as chitin metabolic process and amino sugar metabolic process, which are also enriched among TATA genes (Fig EV1B; Dataset EV6). Furthermore, typical tissue-specific functions were identified among highly paused genes, including rhabdomere development, respiratory system development, and generation of neurons. Although categories such as multicellular organism development and signaling were also enriched, these genes were not well-studied developmental genes.

In summary, while some of these paused genes might be classified as developmental genes by other methods, we conclude that paused promoters clearly contribute to the specific structure and function of differentiated tissues and thus fulfill the criteria for being effector genes. We did however notice differences between the TATA and paused effector genes. The TATA genes were often short genes found in clusters of gene families (e.g., the *Osi* gene family, Appendix Fig S5), and many of them were expressed in tissues such as the epidermis, gut, and trachea, which are exposed to the environment and may require adaptation (Dorer *et al*, 2003; Cornman, 2009; Shah *et al*, 2012). This indicates that effector genes with different promoter types may have different structural and functional properties.

## TATA genes are expressed with high variability but low background expression

We next analyzed whether the different promoter types might display different expression characteristics across tissues. Using the scRNA-seq data, we analyzed their expression noise, measured by the coefficient of variation across different expression bins. We found that TATA genes had consistently higher expression variation than paused genes in all expression bins (Figs 3A and, EV2A and B; Datasets EV4 and EV5).

We then specifically compared the expression variability between TATA and highly paused promoters when the gene was

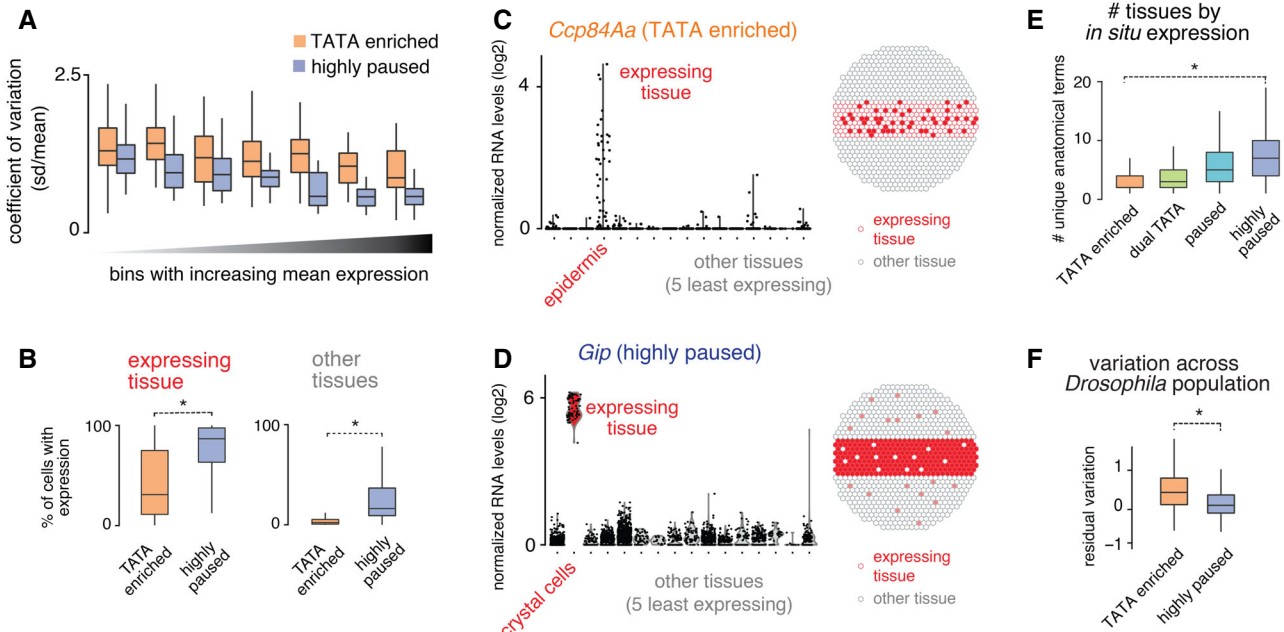

**Figure 3. scRNA-seq reveals differences in expression characteristics of TATA and paused genes.**

A   The differences in cell-to-cell gene expression variability for the TATA and the paused effector gene groups were analyzed using the scRNA-seq data. The coefficient of variation (standard deviation/mean) of gene expression was calculated for all genes in the tissue with the highest expression for each gene. The median coefficient of variation was consistently lower for the paused genes compared with the TATA genes.

B   The frequency of cells with any detectable expression (> one read) was calculated in tissues with the highest expression for each gene (expressing tissue) and in five other tissues with the least expression for each gene (other tissues). The frequency of cells with detectable expression in the expressing tissues, a measure of expression robustness, is lower for the TATA genes compared with the highly paused genes (left) (Wilcoxon two-sided test, $*P < 10^{-15}$). The frequency of cells with detectable expression in the other tissues, a measure of background expression, is also lower for the TATA genes compared with the paused genes (right; Wilcoxon two-sided test, $*P < 10^{-15}$).

C, D   Normalized gene expression levels (read-count normalized for each cell, $\log_2$) in different tissues, from the scRNA-seq experiment, for a TATA group gene, and a highly paused group gene, are shown. (C) The TATA gene, *Ccp84Aa*, shows noisy expression in the epidermis, without detectable background expression in non-expressing tissues. (D) The highly paused gene, *Gip*, shows very robust expression in crystal cells but has high background expression in the non-expressing tissues.

E   The number of annotation terms associated with each gene in the BDGP *in situ* database. This is a measure of whether the expression of a gene is restricted to specific subsets of tissue (Wilcoxon two-sided test, $*P < 10^{-15}$).

F   The coefficient of variation across different isogenic lines of *Drosophila* from DGRP, after being corrected for dependence on expression (loess regression), is plotted for different effector gene groups (Sigalova *et al*, 2020). Genes from the TATA-enriched group show high variability compared with the paused genes (Wilcoxon two-sided test, $*P < 10^{-15}$). Box plots in all panels show the median as the central line, the first and the third quartiles as the box, and the upper and lower whiskers extend from the quartile box to the largest/smallest value within 1.5 times of the interquartile range.

either active in a tissue (expression cluster with highest expression) or inactive (the five expression clusters with least expression). For both the active and inactive expression clusters of each gene, we scored how many cells had detectable expression (at least one read). The results show that paused genes are indeed more consistently expressed between cells when active in a tissue, but they also have higher background expression when not expressed (Figs 3B and EV2F; Dataset EV5). In contrast, TATA genes show higher expression variability when expressed, but less background expression in tissues where the gene is not expressed (Figs 3B and EV2F; Dataset EV5). The difference between promoter types was still present after accounting for differences in their expression levels or gene length (Fig EV2D and E). Good examples to illustrate this fundamental difference are *Ccp84Aa*, a TATA gene expressed in the epidermis (Figs 3C and EV2G for additional examples), and *Gip*, a highly paused gene expressed in crystal cells (Figs 3D and EV2G for additional examples).

The high expression variability of TATA genes has previously been associated with high stochastic noise that is intrinsic to this core promoter type (Raser & O'Shea, 2004; Blake *et al*, 2006; Hornung *et al*, 2012). While our results are consistent with these findings, it is also possible that TATA genes appear to be more variably expressed across cells within a tissue because they are expressed in a more restricted fashion, e.g., in subtypes of cells within a tissue. To test this, we analyzed their *in situ* expression pattern in late-stage *Drosophila* embryos using the BDGP database. We found that the TATA genes had significantly fewer annotated expression patterns compared with highly paused genes (Fig 3E; Dataset EV2). This more restricted expression of TATA genes may therefore also contribute to the increased expression variability and reduced background expression in our scRNA-seq data. Finally, we found that the TATA effector gene group had higher expression variation across *Drosophila* population isolates than the highly paused effector gene group (Figs 3F and EV2C), consistent with TATA promoters being more evolutionarily adaptable.

### TATA promoters have lower chromatin accessibility

We previously observed that paused and TATA promoters in the *Drosophila* embryo have different nucleosome configurations: highly paused promoters have a strong disposition for a promoter nucleosome that is depleted when paused Pol II is present (Gilchrist *et al*, 2010; Gaertner *et al*, 2012; Chen *et al*, 2013), while TATA promoters have fuzzy promoter nucleosomes (Tirosh & Barkai, 2008; Gaertner *et al*, 2012). Consistent with observations in yeast, this suggests a model in which the promoter nucleosome at TATA promoters represents a barrier to transcriptional activation, which in turn may influence its expression characteristics (Raser & O'Shea, 2004; Tirosh & Barkai, 2008; Hornung *et al*, 2012).

We therefore analyzed whether the promoter groups among the effector genes differed in their promoter nucleosome occupancy. Using MNase-seq data, we found that the higher the Pol II penetrance across tissues, the more the promoters became nucleosome depleted in the late embryo as compared to the early embryo, while TATA promoters did not show such change (Figs 4A and EV3A for comparisons with housekeeping and developmental paused genes). This supports the idea that the expression characteristics of the two promoter types involve different nucleosome configurations.

If the promoter nucleosome indeed represents a barrier to activation and this barrier is lowered by the presence of paused Pol II, one would expect that promoters with paused Pol II show higher chromatin accessibility. To test this, we performed ATAC-seq experiments in 14–17 h embryos as a measurement for chromatin accessibility across all tissues (Figs 4B and EV3B; Dataset EV2). Strikingly, we found that the more Pol II pausing, the higher the average promoter accessibility, while the most canonical TATA promoters with the least amount of pausing showed the least chromatin accessibility (Figs 4B and EV3B).

We then tested whether the pattern of chromatin accessibility across promoters was tissue-specific and performed ATAC-seq on tissues isolated using the INTACT method. We found consistently higher promoter accessibility across all tissues for highly paused promoters compared with the TATA promoters (Figs 4C and D, and EV3C; Dataset EV2). This supports our hypothesis that promoters with high levels of Pol II pausing are nucleosome depleted. This in turn could lower the barrier for activation, which explains the robust tissue-specific expression, but also the higher levels of background expression as compared to TATA promoters (Fig 5).

## Discussion

Previous bioinformatics analyses suggested that effector genes are highly enriched among genes with TATA elements (Schug *et al*, 2005; Carninci *et al*, 2006; FitzGerald *et al*, 2006; Engström *et al*, 2007; FANTOM Consortium *et al*, 2014), but how the promoter type affects their tissue-specific expression has not been clear. Moreover, TATA genes have been observed in other contexts to have altered Pol II pausing behavior (Gilchrist *et al*, 2010; Chen *et al*, 2013; Shao & Zeitlinger, 2017), but whether this applies to effector genes in differentiated tissues was not known. Furthermore, it was unclear whether effector genes would be predominantly expressed by TATA promoters or whether other promoter types would also be common for these types of genes. Here, we found that effector genes with TATA promoters are indeed expressed with minimal Pol II pausing, but that there are also many effector genes that are expressed from paused promoters. Since the two promoter types are employed in the same cells, we were able to directly compare the two promoter types side-by-side across different tissues and analyze their tissue-specific expression characteristics using scRNA-seq.

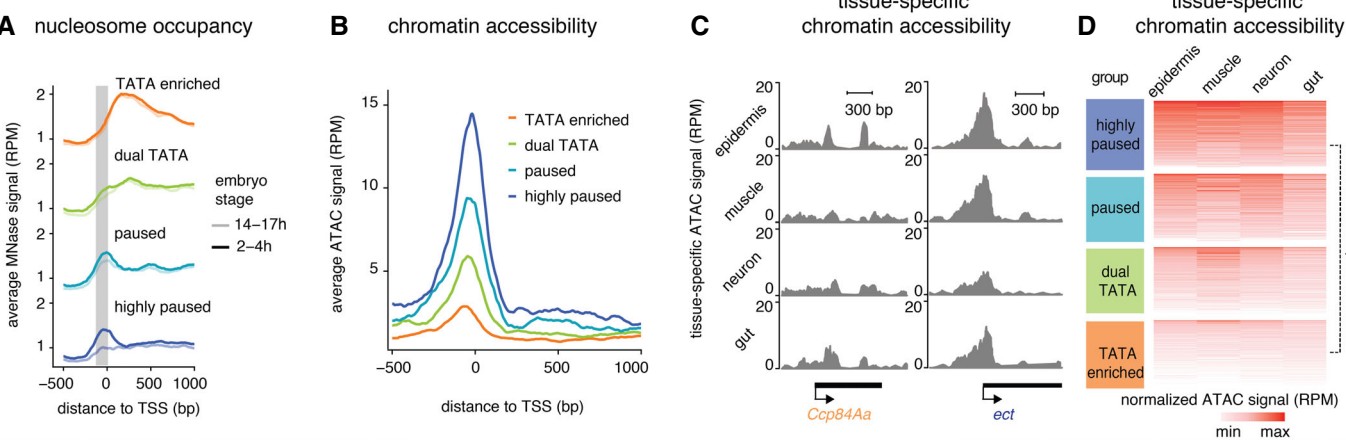

**Figure 4. Paused genes are more accessible than the TATA genes.**

A   Average read-count normalized MNase signal (RPM) from 2–4 h to 14–17 h embryos is shown at the different effector gene promoter groups. The gray area highlights the changes in nucleosome occupancy, which are prominent at highly paused genes.

B   Chromatin accessibility is shown as the average read-count normalized ATAC-seq signal (RPM) from 14–17 h embryos for each promoter group. Paused genes show higher accessibility at the promoter region than the TATA genes.

C, D   ATAC-seq chromatin accessibility was measured in different tissues isolated from 14–17 h embryos using INTACT. (C) Read-count normalized ATAC (RPM) signal at individual genes from the TATA group (*Ccp84Aa*) and highly paused gene group (*ect*) is shown. (D) Read-count normalized ATAC signals (RPM) from different tissues were calculated for each gene from 150 bp upstream of the TSS to the TSS. Paused genes show higher average accessibility across all tissues compared with the TATA genes (Wilcoxon two-sided test, $*P < 2.2*10^{-16}$).

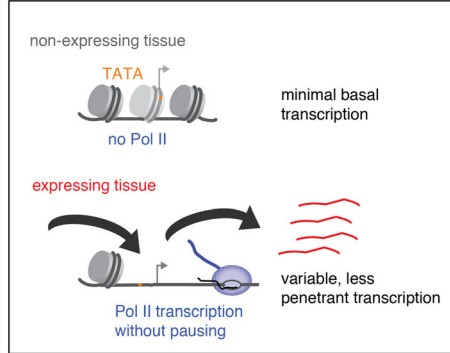

**Figure 5.   Model for the different expression characteristics for paused vs TATA promoters.**
Paused promoters have high levels of paused Pol II throughout the embryo, even in tissues where the genes are not expressed; they show high promoter accessibility in all tissues, low gene expression noise when active, but also high background expression when inactive. TATA promoters mediate highly tissue-restricted expression and only show Pol II occupancy when active; they have lower chromatin accessibility and background expression but show higher expression noise when active.

We found that paused and TATA promoters regulate tissue-specific gene expression in fundamentally different ways. At promoters with pausing elements, paused Pol II is found broadly across tissues, even in tissues where the genes are not expressed. In contrast, TATA promoters only recruit Pol II in tissues where the genes are active and the most canonical TATA promoters do not show Pol II pausing. These patterns of Pol II recruitment across tissues are consistent with those observed over developmental time, since we have previously observed that paused promoters typically show Pol II occupancy prior to induction, while TATA promoters do not (Gaertner *et al*, 2012). Our results therefore consolidate the fundamental difference between these two promoter types.

The difference in Pol II recruitment between promoter types could explain the difference in expression characteristics that we observed in our scRNA-seq data. Paused genes showed significantly lower expression variability within the cells of a tissue as compared to TATA genes. This is consistent with the more synchronous expression of paused genes over time and the generally higher expression variability of TATA genes (Raser & O'Shea, 2004; Tirosh *et al*, 2006; Blake *et al*, 2006; Tirosh & Barkai, 2008; Boettiger & Levine, 2009; Lehner, 2010; Hornung *et al*, 2012; Lagha *et al*, 2013; Li & Gilmour, 2013; Day *et al*, 2016; Faure *et al*, 2017; Sigalova *et al*, 2020). However, paused genes also had higher levels of background expression than TATA genes.

We therefore propose that the presence of paused Pol II, while allowing low variability in gene expression, causes background expression because of the broad recruitment of paused Pol II throughout the embryo. Precise gene expression requires both low variability when the gene is active and low background expression when the gene is inactive. Since neither promoter type fulfills both criteria, there may be a trade-off between them. Highly paused promoters are more optimal for achieving low expression variability when genes are active, while TATA promoters are more optimal for achieving very low background expression when the genes are inactive.

How paused Pol II changes the expression characteristics of promoters is not entirely clear. Since paused Pol II does not directly promote new initiation (Gilchrist *et al*, 2008; Gilchrist *et al*, 2010;

Shao & Zeitlinger, 2017), we favor a model in which paused Pol II lowers the activation barrier for the promoter by keeping the promoter nucleosome away and increasing the accessibility to the promoter (Gilchrist *et al*, 2008; Gilchrist *et al*, 2010). This model is consistent with our analysis of nucleosome occupancy by MNase-seq and chromatin accessibility by ATAC-seq across our promoter types, showing that paused promoters have significantly higher accessibility than TATA promoters.

A significant role for the promoter nucleosome in shaping the expression characteristic of a gene is supported by studies in yeast. TATA promoters are typically occluded by a promoter nucleosome, which is then removed during gene activation (Lee *et al*, 2007; Tirosh *et al*, 2007; Tirosh & Barkai, 2008). The stochastic nature of nucleosome removal has been shown to be associated with gene expression variability (Boeger *et al*, 2008; Kim & O'Shea, 2008; Brown *et al*, 2013; Boeger *et al*, 2015). Paused promoters in *Drosophila*, on the other hand, also have a promoter nucleosome, but this nucleosome is absent when paused Pol II is present (Gilchrist *et al*, 2008; Gilchrist *et al*, 2010; Gaertner *et al*, 2012), which may lower the activation barrier and produce lower gene expression variability.

Other mechanisms could also contribute to the differences in expression variability between the two promoter types. Transcription of many genes is a discontinuous process that occurs in bursts of transcripts (Raj *et al*, 2006; Coulon *et al*, 2013). These bursts of transcription are larger at TATA promoters (Hornung *et al*, 2012; Tantale *et al*, 2016; Larsson *et al*, 2019), presumably because the stable binding of TBP leads to increased reinitiation (Yean & Gralla, 1997; Yean & Gralla, 1999; van Werven *et al*, 2009; Joo *et al*, 2017; Hasegawa & Struhl, 2019). Furthermore, it has been proposed that Pol II pausing could influence the duration of the inactive state and reduce noise (Pedraza & Paulsson, 2008; Suter *et al*, 2011; Boettiger *et al*, 2011; Boettiger, 2013; Zoller *et al*, 2015; Shao *et al*, 2019).

While the expression characteristics can explain why effector genes use different promoter types, a not mutually exclusive possibility is that the two promoter types play different roles in evolution. Effector genes in differentiated tissues are under evolutionary pressure to adapt to a changing environment over time. TATA promoters may be more tunable in their expression levels since they are

more sensitive to mutational perturbations and show higher expression divergence between species (Tirosh *et al*, 2006; Landry *et al*, 2007; Tirosh & Barkai, 2008; Hornung *et al*, 2012; Sigalova *et al*, 2020). Consistent with this hypothesis, we found that TATA effector genes were often short genes in clusters that are expressed in tissues that have to adjust to their environments, such as the epidermis, gut, and trachea. These genes may correspond to the simple "gene batteries" described by Eric Davidson and may be regulated predominantly by promoter-proximal regulatory elements (Erwin & Davidson, 2009; Roider *et al*, 2009; Soler *et al*, 2010). In contrast, promoters with paused Pol II tend to be found in relatively long genes with extensive cis-regulatory regions (Roider *et al*, 2009; Zeitlinger & Stark, 2010; Sigalova *et al*, 2020). A large number of enhancers in these genes makes it more likely that gene duplications disrupt enhancer-promoter interactions and that promoter mutations have detrimental pleiotropic effects.

The characteristics of effector genes that we observed for the late *Drosophila* embryo are likely to be similar in vertebrates. Vertebrate TATA promoters are also enriched among the tissue-specific genes (Schug *et al*, 2005; Carninci *et al*, 2006; FANTOM Consortium *et al*, 2014) and have higher expression variability (Zoller *et al*, 2015; Faure *et al*, 2017; Sigalova *et al*, 2020). Furthermore, the presence of paused Pol II at promoters has been found to correlate with reduced gene expression noise (Day *et al*, 2016). Thus, the possible trade-off between various expression characteristics that we observe at different promoter types could be a general feature of metazoan promoters.

# Materials and Methods

### Fly stocks

Oregon-R embryos were used for wild-type whole-embryo experiments. For the INTACT experiments, embryos from fly stocks expressing tissue-specific RAN-GAP-mcherry-FLAG-BirA were used. To generate these fly stocks, UAS RAN-GAP-mcherry-FLAG-BirA lines were crossed with six tissue-specific Gal4 driver lines with the following Bloomington stock number: neuron - 8760, glia - 7415, trachea - 8807, epidermis - 7021, muscle – 27390, and gut - 110394. The exact genotype is listed in Appendix Table S1.

### Embryo collection

For embryo collections, adult flies were maintained in population cages in an incubator at 25°C. Embryos were collected on apple juice plates and matured in an incubator at 25°C. For example, 14–17 h AED embryos were collected for 3 h and then matured for another 14 h. Embryos were dechorionated for 1 min with 67% bleach and then cross-linked for 15 min with 1.8% formaldehyde (final concentration in water phase). Embryos were flash-frozen in liquid nitrogen and stored at −80°C. For ATAC-seq and scRNA-seq experiments, the embryos were not cross-linked and processed immediately after dechorionation. Experiments were performed in biological replicates from independent embryo collections on different days. At least two biological replicates were performed for scRNA-seq, ATAC-seq, ChIP-seq, and bulk RNA-seq experiments. One or two replicates were performed for the MNAse-seq experiments.

### Isolation of tissue-specific nuclei for ChIP-seq or ATAC-seq

Nuclei isolation was performed using previously published protocols with modifications (Deal & Henikoff, 2011; Bonn *et al*, 2012). 0.5 g of embryos were dounced in HBS buffer (0.125 M Sucrose, 15 mM Tris [pH 7.5], 15 mM NaCl, 40 mM KCl, 2 mM EDTA, 0.5 mM EGTA, 2% BSA, protease inhibitors) in a 15 ml dounce tissue grinder. The nuclei suspension was then filtered through two layers of miracloth (Calbiochem, #475855) and spun at 500 g for 10 min at 4°C. The supernatant was discarded. The nuclear pellet was resuspended in HBS buffer and dissociated by passage through a syringe (22.5-gauge needle) 10 times. After spinning again, the pellet was resuspended in HBS buffer and incubated with Dynabeads® M-280 Streptavidin beads (Invitrogen, # 11205D) for 30 min with end-to-end rotation at 4°C. A magnet was used to separate the bead-bound nuclei, and the beads were washed thoroughly with HBS buffer. For tissue-specific ChIP-seq experiments, multiple batches are typically combined, and each ChIP is performed on ~ 0.5–1 g starting embryos containing ~ 5 μg DNA based on measuring the chromatin by Nanodrop or Qubit. The NEBNext ChIP-Seq Library Prep kit was used for library preparation. For tissue-specific ATAC-seq, much smaller amounts are used as starting material (e.g., 2,000 embryos).

### ATAC-seq experiments

Oregon-R embryos of stage 14–17 h AED were dounced in HBS buffer as described above, starting with ~ 500 embryos for whole-embryo ATAC-seq or ~ 2,000 embryos for tissue-specific ATAC-seq. The transposition of the nuclei was performed as described in (Buenrostro *et al*, 2013), using 2.5 μl Tn5 transposase, followed by PCR amplification (Nextera DNA Sample Preparation Kit: FC-121-1030, Illumina) and library preparation (the Nextera index kit: FC-121-1011, Illumina). Libraries were purified using Agencourt AMPure XP beads (A63881, Beckman Coulter). Paired-end sequencing was performed on the NextSeq 500 instrument (Illumina). Following sequencing, the chromatin accessibility was calculated by computationally filtering for fragments of sizes up to 100 bp, which represent small fragments from accessible regions.

### MNase-seq experiments

MNase digestion was performed similarly to previously published protocols (Chen *et al*, 2013). Briefly, chromatin was extracted from 0.1 g of Oregon-R embryos per replicate and then digested with a concentration gradient of MNase (Worthington Biochemical Corporation #LS004798) for 30 min at 37°C. All samples were run on a gel, and the digestion concentration to be sequenced was chosen as previously described (Chen *et al*, 2013). Libraries were prepared using the NEBNext DNA Library Prep Kit and then paired-end sequenced with an Illumina HiSeq 2500 sequencing system. The nucleosome-sized fragments (100–200 bp) were selected computationally to analyze the nucleosome occupancy.

### mRNA-seq experiments

Total mRNA was extracted from non-cross-linked embryos using the Maxwell Total mRNA purification kit (Promega, #AS1225).

PolyA-mRNA was isolated using DynaI oligo(dT) beads (Life Technologies, #61002). Libraries were prepared using the TruSeq DNA Sample Preparation Kit (Illumina, #FC-121-2001) and sequenced on the HiSeq 2500 and the Nextseq 500 (Illumina).

### scRNA-seq experiments

scRNA-seq experiments were performed on 14–14.5 h AED wild-type Oregon-R embryos. The isolation of single cells was performed similarly to the previously published protocol (Karaiskos *et al*, 2017) with the following modifications. The embryos were dounced in SFX medium with 0.1% PF68 + 0.1% BSA, which was found to improve the cell viability. The total number of dounces was increased to 120 to improve the isolation of cells from late-stage embryos. Isolated cells were filtered and washed and then resuspended in Schneider's medium to avoid any interference with droplet formations in subsequent steps. Resuspended cells were immediately processed in the 10x Genomics instrument with optimal loading at a targeted capture rate of about 6,000 cells per run to minimize doublets. RNA isolation and cDNA synthesis and amplification were done according to the manufacturer's instructions. Libraries were prepared using the TruSeq DNA Sample Preparation Kit (Illumina, #FC-121-2001) and sequenced on the HiSeq 2000. scRNA-seq experiments were performed on two biological replicates on separate days from different cages.

### Sequence alignment

All sequencing reads were aligned to the *Drosophila melanogaster* genome (dm6) using Bowtie (v 1.1.2) (Langmead *et al*, 2009), allowing a maximum of two mismatches and including only uniquely aligned reads. The sequenced reads were trimmed to 50 bp before alignment. Aligned reads were then extended to the estimated insert size or the actual size for the paired-end libraries. For the bulk mRNA-seq samples, the gene expression values were calculated by performing pseudo-alignment using the Kallisto package (Bray *et al*, 2016). For the scRNA-seq samples, alignment and separations of reads from different cells were done using the Cell Ranger pipeline (v 2.1.1) from 10× Genomics.

### Reannotation of TSS using CAGE data

For general purposes, we used the flybase gene annotation (r 6.21) for our analysis. For the motif enrichment analysis, sequence heat map, sequence logos, ATAC-seq, and MNAse-seq metagene plots, we reannotated the existing TSSs based on CAGE data obtained from 16–18 h embryos (modENCODE - ID:5344). A flybase TSS was reannotated based on the highest signal found within 150 bp around the TSS.

### Mapping scRNA-seq data to known tissue types

We obtained the gene expression profiles of about 3,500 cells in total from two independent biological replicates. The cells from both replicates were pooled for the downstream analysis. The Seurat package (Satija *et al*, 2015) was used for normalization, clustering, and visualization of the scRNA-seq data. The cell gene expression matrix was normalized by the total expression per cell and scaled by a factor of 10,000 and log-transformed. Principal component analysis was performed on highly variable genes. The first 20 principal components were used as input for clustering by Shared Nearest Neighbor method. The Seurat package was used to identify the marker genes for each of the clusters.

The tissue of origin for the clusters was identified by comparing the scRNA-seq expression patterns with the *in situ* hybridization profiles from the Berkeley *Drosophila* Genome Project (BDGP) similar to the previously published method (Karaiskos *et al*, 2017). Briefly, the annotated gene expression profiles for the embryonic stage 13–16 were obtained from BDGP, excluding the ubiquitously expressed genes. The scRNA-seq data were then binarized into ON/OFF values, based on whether the expression values are above/below a threshold. The expression value at the 0.9 quantile for each gene was used as the threshold above, which it was considered ON. The results did not vary significantly for a wide range of cutoffs. The Matthews Correlation Coefficient was calculated based on this binarized version of our data vs the binarized BDGP data. Each cell was annotated as the tissue with which it had the maximum correlation. Each cluster was then assigned the tissue to which the largest number of cells were annotated. For ambiguous clusters, the occurrence of known tissue markers was analyzed and clusters were manually merged or separated such that they better matched anatomical structures. For example, when more than one cluster was annotated with the same tissue type and we could not find meaningful differences between them, we merged these clusters. Similarly, when small subgroups with distinct tissue types were found within a cluster, the clusters were separated into multiple sub-clusters.

### Gene groups

We defined effector genes by their late upregulation during embryogenesis (> 5×, $P < 0.05$ from 2–4 h to 14–17 h [Wald test, Deseq2 library], > 10 TPM in 14–17 h, < 2 TPM in 2–4 h) which yielded 1,527 genes. As control groups, we also defined ubiquitously expressed housekeeping genes ($P > 0.05$ from 2–4 h to 14–17 h [Wald test, Deseq2 library], > 10 TPM in 14–17 h, > 10 TPM in 2–4 h; 647 genes), as well as developmental genes that are highly paused throughout embryogenesis (772 genes) (Gaertner *et al*, 2012) (Datasets EV1 and EV2). We later grouped the effector genes based on Pol II penetrance, defined as the number of tissues (from 0 to 6 tissues) in which Pol II is detected around the transcription start site (TSS to 200 bp downstream of the TSS) to be above background (> 2-fold signal over input). This resulted in four groups: TATA enriched (527 genes)(0 tissues), dual TATA (415 genes)(1–2 tissues), paused (222 genes)(3–4 tissues), and highly paused (362 genes)(5–6 tissues).

### Promoter element enrichment

In Fig 2C and EV1C, Appendix Fig S5B, the presence of known *Drosophila* promoter elements in each promoter is identified with zero mismatches, in a specified window relative to the TSS (Appendix Table S2). For each gene group and each promoter element, the enrichment was calculated as the fraction of genes in a group with a promoter element over the fraction of all annotated genes with the same promoter element. For genes with multiple isoforms, the isoform with the highest Pol II signal in 14–17 h

embryos was used for the analysis. The statistical significance was calculated with the Fisher's exact test after correcting for multiple testing by the Benjamini–Hochberg method.

### Sequence heat map and logo plots

Sequence heat maps were made by plotting the nucleotides at each position using the ggplot2 package (geom_tile). We randomly selected a subset of genes (350) for each group so that the plots are visually comparable. Logo plots are made using the ggseqlogo package.

### Pausing index calculations

Pausing index in Fig EV1D and G was calculated as the amount of Pol II ChIP-seq signal in the 200 bp window downstream of the TSS divided by the Pol II signal in the 200–400 bp region downstream from the TSS in the gene body.

### % of cells with expression and coefficient of variation calculations

When calculating the percentage of cells with detectable transcripts for each gene in each tissue, the tissue with maximum expression was considered as the expressing tissue and the five least expressing tissues were considered as other tissues. Cells with at least one detectable read are considered as expressing cells. The coefficient of variation (CV) was calculated as the ratio of the standard deviation of expression divided by the mean expression in the expressing tissue. Only the cells with detectable transcripts were considered for this calculation. The CV exhibited a strong negative relationship with mean expression level. To account for this relationship, we applied local polynomial regression (loess regression, degree = 2, span = 0.1) and calculated the residual coefficient of variation from the fitted line.

### Statistical significance calculations and data visualization

*P* values in Figs 3B, E and F, 4D, EV1D and G, EV2B, C and F, and EV3C, Appendix Fig S4A and B, and S5C were calculated with the two-sided Wilcoxon test. *P* values in Figs 2C and EV1C, and Appendix Fig S5B were calculated with the Fisher's exact test with multiple-testing correction, $*P < 0.05$. *P* values in Appendix Fig EV1A and B were calculated using the hypergeometric test. Heat maps are normalized, and really low or high values are ceiled or floored, respectively. Box plots show the median as the central line, the first and the third quartiles as the box, and the upper and lower whiskers extend from the quartile box to the largest/smallest value within 1.5 times of the interquartile range.

## Data and software availability

Raw and processed data associated with this manuscript have been deposited in GEO under the accession number GSE120157 (https://www.ncbi.nlm.nih.gov/geo/query/acc.cgi?acc=GSE120157). All data analyses performed in this paper, including raw data, processed data, software tools, and analysis scripts are available through a publicly accessible Amazon machine image (ami-id: ami-0054641ba9378d685). The analysis code is also available on GitHub at https://github.com/zeitlingerlab/Ramalingam_promoter_types_2020.

**Expanded View** for this article is available online.

## Acknowledgements

We thank Dr. Steve Henikoff for sharing the $w^{1118}$; p[UASRG]5/CyO, p[twi-GAL4] p [UAS-EGFP] and $w^{1118}$; p[UASRG]6 fly stocks, which were used to generate the tissue-specific tagged RAN-GAP expressing fly lines. We thank Kate Hall, Allison Peak, and Ana Pinson for technical assistance with the scRNA-seq experiments. We thank Mark Miller for help with illustrations. We thank Robb Krumlauf, Mounia Lagha, Viraj Doddihal, Zainab Afzal, Charles McAnany, Kaelan Brennan, Curtis Bacon, and Melanie Weilert for their feedback on the manuscript. This work was supported by funding from the National Institutes of Health (DP2OD004561) and the Stowers Institute for Medical Research.

## Author contributions

MN, VR, and JZ conceived the study and designed the experiments. MN and VR carried out the INTACT experiments. VR performed the scRNA-seq experiments. VR and JJ analyzed the data. VR, JZ, MN, and JJ contributed to data interpretation. VR and JZ wrote the manuscript.

## Conflict of interest

The authors declare that they have no conflict of interest.

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
