## [Review Process File · Molecular Systems Biology]

TATA and paused promoters active in differentiated tissues have distinct expression characteristics

Vivekanandan Ramalingam, Malini Natarajan, Jeff Johnston, and Julia Zeitlinger
DOI: [10.15252/msb.20209866](https://doi.org/10.15252/msb.20209866)

Corresponding author: Julia Zeitlinger (jhz@stowers.org)

Review Timeline:

Submission Date:	17th Jul 20
Editorial Decision:	20th Aug 20
Revision Received:	29th Oct 20
Editorial Decision:	14th Dec 20
Revision Received:	22nd Dec 20
Accepted:	7th Jan 21

Editor: Jingyi Hou

Transaction Report:

20th Aug 2020

Manuscript Number: MSB-20-9866

Title: TATA and paused promoters active in differentiated tissues have distinct expression characteristics

Author: Vivekanandan Ramalingam

Malini Natarajan

Jeff Johnston

Julia Zeitlinger

Thank you for submitting your work to Molecular Systems Biology. We have now heard back from the three reviewers who agreed to evaluate your manuscript. As you will see below, the reviewers think the study is interesting. They raise however a series of concerns, which we would ask you to address in a major revision.

I think that the reviewers' recommendations are rather clear and there is no need to reiterate their comments. In particular, some of the analyses would need to be extended to the developmental genes as suggested by reviewers #1 and #2. As you may already know, our editorial policy allows in principle a single round of major revision, so it is essential to provide responses to the reviewers' comments that are as complete as possible. Please feel free to contact me in case you would like to discuss in further detail any of the issues raised by the reviewers.

On a more editorial level, please do the following:

- Please provide a .docx formatted version of the manuscript text (including legends for main figures, EV figures and tables). Please make sure that the changes are highlighted to be clearly visible.

- Please provide individual production quality figure files as .eps, .tif, .jpg (one file per figure).

-Please provide a .docx formatted letter INCLUDING the reviewers' reports and your detailed point-by-point responses to their comments. As part of the EMBO Press transparent editorial process, the point-by-point response is part of the Review Process File (RPF), which will be published alongside your paper.

-Please note that all corresponding authors are required to supply an ORCID ID for their name upon submission of a revised manuscript.

-We replaced Supplementary Information with Expanded View (EV) Figures and Tables that are collapsible/expandable online (see examples in <http://msb.embopress.org/content/11/6/812>). A maximum of 5 EV Figures can be typeset. EV Figures should be cited as 'Figure EV1, Figure EV2' etc... in the text and their respective legends should be included in the main text after the legends of regular figures.

Additional Tables/Datasets should be labeled and referred to as Table EV1, Dataset EV1, etc. Legends have to be provided in a separate tab in case of .xls files. Alternatively, the legend can be

supplied as a separate text file (README) and zipped together with the Table/Dataset file.

For the figures and tables that you do NOT wish to display as Expanded View figures, they should be bundled together with their legends in a single PDF file called *Appendix*, which should start with a short Table of Content. Each legend should be below the corresponding Figure/Table in the Appendix. Appendix figures and tables should be referred to in the main text as: "Appendix Figure S1, Appendix Figure S2, Appendix Table S1" etc. See detailed instructions regarding expanded view here: <https://www.embopress.org/page/journal/17444292/authorguide#expandedview>.

-Before submitting your revision, primary datasets (and computer code, where appropriate) produced in this study need to be deposited in an appropriate public database (see <https://www.embopress.org/page/journal/17444292/authorguide#dataavailability>).

The accession numbers and database should be listed in a formal "Data Availability " section (placed after Materials & Method) that follows the model below (see also <https://www.embopress.org/page/journal/17444292/authorguide#dataavailability>). Please note that the Data Availability Section is restricted to new primary data that are part of this study.

Data availability

- We would encourage you to include the source data for figure panels that show essential quantitative information. Additional information on source data and instruction on how to label the files are available at < <https://www.embopress.org/page/journal/17444292/authorguide#sourcedata> >.

- All Materials and Methods need to be described in the main text. We would encourage you to use 'Structured Methods', our new Materials and Methods format. According to this format, the Material and Methods section should include a Reagents and Tools Table (listing key reagents, experimental models, software and relevant equipment and including their sources and relevant identifiers) followed by a Methods and Protocols section in which we encourage the authors to describe their methods using a step-by-step protocol format with bullet points, to facilitate the adoption of the methodologies across labs. More information on how to adhere to this format as well as downloadable templates (.doc or .xls) for the Reagents and Tools Table can be found in our author guidelines: <

<https://www.embopress.org/page/journal/17444292/authorguide#researcharticleguide>>. An example of a Method paper with Structured Methods can be found here: .

- Regarding data quantification:

Please ensure to specify the name of the statistical test used to generate error bars and P values,

the number (n) of independent experiments (please specify technical or biological replicates) underlying each data point and the test used to calculate p-values in each figure legend. Discussion of statistical methodology can be reported in the materials and methods section, but figure legends should contain a basic description of n, P and the test applied.

Graphs must include a description of the bars and the error bars (s.d., s.e.m.).

- Please provide a "standfirst text" summarizing the study in one or two sentences (approximately 250 characters, including space), three to four "bullet points" highlighting the main findings and a "synopsis image" (550px width and max 400px height, jpeg format) to highlight the paper on our homepage.

Here are a couple of examples:

<https://www.embopress.org/doi/10.15252/msb.20199356>

<https://www.embopress.org/doi/10.15252/msb.20209475>

<https://www.embopress.org/doi/10.15252/msb.209495>

When you resubmit your manuscript, please download our CHECKLIST

(<http://bit.ly/EMBOPressAuthorChecklist>) and include the completed form in your submission.

Please note that the Author Checklist will be published alongside the paper as part of the transparent process

(<https://www.embopress.org/page/journal/17444292/authorguide#transparentprocess>).

If you feel you can satisfactorily deal with these points and those listed by the referees, you may wish to submit a revised version of your manuscript. Please attach a covering letter giving details of the way in which you have handled each of the points raised by the referees. A revised manuscript will be once again subject to review and you probably understand that we can give you no guarantee at this stage that the eventual outcome will be favorable.

Yours sincerely,

Jingyi Hou

Editor

Molecular Systems Biology

If you do choose to resubmit, please click on the link below to submit the revision online *within 90 days*.

Link Not Available

IMPORTANT: When you send your revision, we will require the following items:

1. the manuscript text in LaTeX, RTF or MS Word format
2. a letter with a detailed description of the changes made in response to the referees. Please specify clearly the exact places in the text (pages and paragraphs) where each change has been made in response to each specific comment given
3. three to four 'bullet points' highlighting the main findings of your study
4. a short 'blurb' text summarizing in two sentences the study (max. 250 characters)
5. a 'thumbnail image' (550px width and max 400px height, Illustrator, PowerPoint or jpeg format),

which can be used as 'visual title' for the synopsis section of your paper.

6. Please include an author contributions statement after the Acknowledgements section (see <https://www.embopress.org/page/journal/17444292/authorguide>)

7. Please complete the CHECKLIST available at (<http://bit.ly/EMBOPressAuthorChecklist>). Please note that the Author Checklist will be published alongside the paper as part of the transparent process

(<https://www.embopress.org/page/journal/17444292/authorguide#transparentprocess>).

8. Please note that corresponding authors are required to supply an ORCID ID for their name upon submission of a revised manuscript (EMBO Press signed a joint statement to encourage ORCID adoption). (<https://www.embopress.org/page/journal/17444292/authorguide#editorialprocess>)

Currently, our records indicate that there is no ORCID associated with your account.

Please click the link below to provide an ORCID:

Link Not Available

The system will prompt you to fill in your funding and payment information. This will allow Wiley to send you a quote for the article processing charge (APC) in case of acceptance. This quote takes into account any reduction or fee waivers that you may be eligible for. Authors do not need to pay any fees before their manuscript is accepted and transferred to the publisher.

*** PLEASE NOTE *** As part of the EMBO Press transparent editorial process initiative (see our Editorial at <http://dx.doi.org/10.1038/msb.2010.72>), Molecular Systems Biology publishes online a Review Process File with each accepted manuscripts. This file will be published in conjunction with your paper and will include the anonymous referee reports, your point-by-point response and all pertinent correspondence relating to the manuscript. If you do NOT want this File to be published, please inform the editorial office at msb@embo.org within 14 days upon receipt of the present letter.

Reviewer #1:

In the manuscript "TATA and paused promoters active in differentiated tissues have distinct expression characteristics", Ramalingam et al characterize expression properties of different promoter types of effector genes in late Drosophila embryo using scRNA-seq combined with profiling of polymerase pausing, chromatin accessibility and nucleosome occupancy. The study is important because it shows different modes of regulation of effector genes (genes responsible for the structure and function of differentiated tissues), which have been previously mostly linked to TATA-box promoters. In particular, authors propose that "highly paused promoters are more optimal for achieving low expression variability when genes are active, while TATA promoters are more optimal for achieving very low background expression when the genes are inactive". The manuscript is well-written, and results are supported by data. Yet, in parts, it is was not clear what the exact analyses where that the authors performed. See specific comments below. The main novelty of this work is the description, characterization and function of TATA-box and PolII-pausing promoters in tissue-specific effector genes, thus excluding developmental and house-keeping

genes. However, since much is known about the latter, especially in the context of TATA and PolII, the study would benefit from a broader perspective by including these types of promoters as separate classes in their computational analysis where possible, to put their findings into a broader context. See more specific comments below:

Major comments:

1. One of the main conclusions of the study is that there are two fundamentally different modes of regulation within effector genes, as reflected by differences in polymerase pausing, nucleosome occupancy and expression noise of these genes. Also, the authors propose in the Discussion that "the two promoter types play different roles in evolution". Given these conclusions, the study would benefit from more in-depth characterization of functional differences between these different types of effector genes.

2. It was not clear what set of genes the authors actually take into account for their analyses - this reviewer was not familiar with the term 'effector genes'. They should provide their definition of "effector genes" and they should give numbers of how many genes these are the first time they talk about it (i.e. p2).

3. More generally, the authors should consider including data for housekeeping and developmental genes (e.g. as background or comparison to effector genes) to put their findings into a broader context. This could make the results for effector genes stronger and also highlight novelty of the study. If not, at least it has to be more clearly stated, that only a subset of genes is analyzed (and number have to be given).

Some interesting questions along these lines are:

- a. Strong pausing was previously linked to developmental genes, while effector genes are mostly associated with TATA-box. Is there a difference in regulation / promoter architecture of highly paused effector vs developmental genes?
- b. GAF (GAGA factor) has been previously linked to PolII pausing in *Drosophila* (Tsay et al 2016, PMID: 27468311). Here, the authors found the GAGA motif moderately enriched in all gene groups. What was the background for this enrichment? Does this imply that effector genes don't depend on GAF for pausing and there is an additional pausing factor for effector genes? Would the same enrichments be observed if GAF ChIP-seq was used in addition to GAGA-motif (since there is at least one other TF CLAMP known to bind GAGA-motif in promoters)? Can the authors distinguish "developmental pausing" from "effector pausing" by the GATA motif? Here a comparison with developmental genes would be helpful.
- c. Li & Gilmour 2013 (PMID 23708796) propose two pausing mechanisms for M1BP-bound genes (mostly, housekeeping genes, transient pausing) and GAF-bound genes (mostly, developmental genes, strong pausing). Also, authors report similar results regarding pausing, nucleosome occupancy and expression variability for highly paused vs. TATA-box genes. How do gene groups identified here agree with those in (Li&Gilmour 2013)? The authors should cite this paper.

4. Can the authors exclude the that the pausing group of effector genes simply arose from an incomplete removal of developmental genes? (Sup Fig 3B shows Development as GO term in paused genes)

5. Figure 3 presents some of the key results, and the authors should show more convincing examples in panel D such that the genes in the two classes have a similar mean expression level (the same way the genes were grouped by mean expression in panel A). In the current examples it

is difficult to judge whether the background in the TATA genes is really different from the background in the pausing genes. Maybe the authors can pick 3 genes from three bins of expression level (low, medium high) to show this is true for all expression levels.

In addition the authors should visualize some aggregate information about background expression levels across genes (paused vs tata-box). The percentage of cells with expression shows only noise not really background expression levels.

6. The methods description and/or figure legends miss some details regarding the analysis, data sources, sample sizes/significance for some group comparisons, which should be clarified in the main text and / or methods section. In particular:

- How are housekeeping and developmental genes defined (sources)? The statement: "By eliminating developmental genes and ubiquitously-expressed housekeeping genes from all late-expressed genes" has to be accompanied by some reference / or the actual list of genes that were removed.
- Which genes are used as background for GO enrichments?
- For the PolII tissue-specificity statement: "The Pol II occupancy was however not always tissue-specific. Some genes showed high Pol II occupancy at the promoter in many or all tissues, despite being expressed in a very tissue-restricted fashion." How many genes are we talking about in each class? Please cite the numbers in the text. In general, it would be appreciated to cite more numbers in the text.
- Fig 2c - the grouping is misleading, it seems the promoters are grouped by #of tissues with PolII enrichments, but the last column indicates this exactly corresponds to the TATA / pausing etc classification. In the text they mention it is enrichment, so they should show it as enrichment, rather than just colors. Also, what exactly is then the colors in Fig 2D/E corresponding to? - the actual pausing/ TATA classes, or the #tissues with PolII enrichment? Is it only one promoter group enriched for each of the 7 classes? If not, maybe some more quantitative visualisation could be helpful (e.g. groups in columns showing enrichment in rows - same as for promoter motifs).
- Fig2d - it is not clear what expression data is shown. Please clarify in the legend.
- Fig2e - data shown on the x and y-axes seem to be redundant since it's Pausing index on y-axis and groups based on pausing on the x-axis. Might be more informative to show the 7 groups from 2c (based on the number of tissues with PolII enrichment) on the x-axis.
- What are number of genes and significance of mean difference in panel 3a?
- Gene expression noise (referring to Fig3a) - this metric is usually calculated by adjusting coefficient of variation for expression level since there is strong negative relationship between CV and expression level (this is seen for highly paused genes). While probably results in panel 3a won't be affected, this adjustment would allow for more accurate comparisons of noise levels among gene groups (e.g. for estimating statistical significance of differences in noise between paused and TATA genes).

Minor points:

- Is Figure 1B really the best visualization for the single cell data? It is somewhat difficult to spot the data within all the lines and the schematics.
- Fig3b,c - what was the threshold on expression level to define expressing cells?
- Fig4a - are those genes expressed at both time-points (2-4 and 14-17h)? Is the difference in accessibility significant for dual TATA and paused genes? Please clarify in the text.
- p5: "We found that the TATA genes had indeed significantly fewer annotated expression patterns compared to..." - is the end of the sentence missing? Shall this refer to panel 3F?
- p2: header should specify that this is effector genes

Reviewer #2:

The manuscript by Ramalingam et al presents evidence that TATA-dependent tissue-specific promoters in *Drosophila* have distinct properties from TATA-less tissue-dependent promoters with respect to PolII pausing, cell-to-cell variability of expression levels, and background "leaky" expression in the cells and tissues in which the gene is (mostly) inactive. They used single cell RNA-seq to determine tissue specificity, variability and background levels of expression of individual genes, and tissue-specific PolII ChIP-seq to determine PolII pausing behaviour at each.

This is a very nice body of work that shows exactly what it claims it is showing, using elegant, cutting-edge methodology, and it should be of great interest to others in the field. What it would benefit from is better integration with what is already known about the differences between TATA-dependent and TATA-independent promoters, both in *Drosophila* and in other organisms:

- The authors removed "developmental" promoters to compare TATA-dependent and TATA-independent "late" tissue-specific promoters, and found differences. I believe that there is a flaw here. DPE- and pause button-dependent promoters differ from TATA-dependent promoters primarily with regard to their dependence on long-range enhancers. In that sense, the TATA-less, DPE- or pause-button containing tissue-specific promoters will probably be a part of the continuum of developmental promoters - just those whose associated enhancers are active only relatively late in development and differentiation, and in a small number of tissues. Other than that, I would expect their properties to be similar to, or even indistinguishable from, developmental promoters active early in development or in multiple tissues. For that reason, it would be essential to extend the analysis to the removed developmental promoters to examine this relationship.

- Indeed, even though the authors claim that there is no clear difference in GO terms between TATA and highly paused genes, it is no surprise to me at all that "developmental process" and "cell communication", both terms known to be associated with multicellular processes under long-term enhancer regulation, show up with the highly-paused genes. If more highly paused genes are included, the overrepresentation of the two terms will only become stronger.

- The difference with respect to the dependence on long-range enhancers will have several consequences that the author observe, and are expected: in a secondarily compacted genome such as that of *Drosophila*, the genes with TATA-dependent promoters will have shorter introns, and more likely to occur as tandem duplicates. On the other hand, DPE-dependent promoters, as essentially developmental promoters often controlled by multiple enhancers, will often have longer introns and intergenic spaces needed to accommodate those enhancers, and will be more dosage-sensitive, both of which would make tandem duplication disruptive with respect to enhancer-promoter interactions.

- Some previous research suggests that, unlike developmental promoters, most TATA-dependent promoters in vertebrates are actually regulated by proximal cis-regulatory modules just upstream of the core promoters (e.g. Roeder HG et al NAR 2009 showed that the difference between TATA-dependent and TATA-independent liver- and muscle-specific genes is that the non-CpG, TATA-dependent ones are the only ones that contain an enrichment of tissue-specific motif at the proximal promoters; Soler E et al Genes Dev 2010 show a rather dramatic difference in the relative position of GATA1 binding between the two types of erythroid-constrained promoters in mouse).

This would suggest the difference in mechanism between TATA-dependent and developmental, enhancer-driven promoters: the former are fully inactive until a context-specific transcription factor binds to them; the latter are poised to respond to enhancer activation, which may be accomplished by starting the transcription and pausing it until enhancer interaction releases the pause.

- The chromatin accessibility of TATA promoters will be influenced by their two properties that are already well known: 1) the nucleosome positions at TATA promoters are not as precise as at most non-TATA promoters, because it is the TATA box position that directs the TSS selection to about 30 bp downstream of it, regardless of the stable position of the nucleosomes at each particular promoter (see e.g. Rach E et al PLOS Genet 2011, and a detailed analysis in Dreos R et al PLOS Comp Biol 2016). It might be that the positions are relatively stable at individual promoters, but since they are not at a constant distance from the TSS, there is no reference they line up with in heatmaps or metaplots. Alternatively, since TATA-dependent promoters act in large "bursts" with periods of inactivity between the bursts, the nucleosome arrangement will be a time average between active and inactive states.

- More generally, what is missing is the analysis of additional promoter features that are known to be associated with different promoter types. It is unclear how the authors defined transcription start sites, but defining them by high-quality CAGE data (from modENCODE or from Eileen Furlong's lab) is more likely to reveal precise spatial dependencies between transcription start sites and motifs such as TATA-box and DPE, as well as MNase signal and any nucleosome positioning signals in the sequence, if present. Also, the authors do not mention if the analysed promoters are predominantly "peaked" (single TSS position) or "dispersed" (multiple TSS positions within the same promoter region). TATA- and DPE-dependent promoters are generally peaked with constrained spacing between the TSS and the motif (e.g. Dreos R PLOS Comp Biol 2016).

- Currently, there is not enough evidence to support the author's claim that accessibility does not change between expressing and non-expressing tissues as there is no direct comparisons but rather very global trends. It would be interesting to directly compare accessibility in expressing and non-expressing tissues for the same genes, as heatmaps with genes separated into their tissue of maximum expression.

Minor:

- It would be helpful if the authors could clearly define Pol II penetrance in the main text, how it is calculated and how it differs from the pausing index it is compared to.

- The motif identification allowing zero mismatches is quite stringent. Would the enrichments apply if lower thresholds were applied, for example 90% match to the PWM?

- In Figure 3 it is not immediately clear what the threshold for expression is for a cell to be classified as expressing.

- In figure 4D it would be better if the heatmaps were ordered by accessibility (highly accessible regions at the top, low accessibility regions at the bottom). It would be easier to assess whether higher accessibility is a global feature of these promoters or if few regions with very high accessibility are skewing the trend.

- In Figure S4, can the authors comment on why we do not see highly paused promoters reflected here, with consistent Pol II occupancy in all tissues?

- In supplementary excel 3, it is not clear what is denoted in new_start and new_end columns.

Reviewer #3:

This paper describes the relationship between promoter types and expression features in the late *Drosophila* embryo. First, single cell RNA-seq (scRNA-seq; with cells derived from whole embryos) was used to obtain gene expression data in various tissues (Fig. 1). Then, by using tissue-specific ChIP-seq with six different tissues, the Pol II occupancy in different tissues was determined (Fig. 2). Genes with high Pol II 'penetrance' (i.e., widespread Pol II occupancy in different tissues) were found to be noisier when inactive and less variable when active, relative to genes with low Pol II penetrance (Fig. 3). Chromatin accessibility also correlates with Pol II penetrance (Fig. 4).

This manuscript is clearly written, and the experiments were well designed. If the Major comment is addressed appropriately, publication would be recommended.

Major comment:

1. In Fig. 2C, the authors classify genes based on the occurrence of enriched Pol II (from the +1 TSS to +200) in different tissues. Accordingly, genes in which Pol II is enriched in 5/6 or 6/6 tissues are called "highly paused". In contrast, genes in which Pol II is not enriched (in 0/6 tissues) are called "TATA". In the "TATA" genes, the TATA-box appears to be enriched by about 4 fold; however, it is likely that many of the "TATA" genes lack a TATA box. Therefore, unless all of the "TATA" genes contain a TATA box (and all of the "Highly Paused" genes lack a TATA box), the "TATA" gene category should be re-named because the name would be inaccurate and misleading. One possible alternate name would be "Non-paused", as used in Fig. 2B. This name would match the basis upon which the genes were categorized. It could be stated that the "Non-paused" genes are enriched for TATA boxes.

Other comments:

2. Please specify exactly how the scRNA-seq data were linked to the tissue-specific Pol II occupancy. It is not obvious that there is a good match between the tissues in the two different experiments. No correlation value is shown in Fig. S4, which could help establish the link. In addition, the data presented in Fig. S4 are restricted to genes with Pol II occupancy in a single tissue.

3. In Fig. 3B-C, it would be useful to show the distribution of normalized RNA levels in addition to the percent of expressing cells. For example, a 2D plot of Normalized RNA levels vs. % Cells with Expression might be informative. "Cells with expression" should also be clearly defined [there is a similar issue in Fig. 2 for "(non-)expressing tissues"].

4. In the legend of Fig. 2, it is stated that the differences in paused Pol II occupancy are related to sample preparation and transcription. Please explain.

We thank the reviewers for their insightful and constructive criticisms and suggestions, which led us to perform additional analyses and improve the overall clarity of the revised manuscript. First, we clarified and validated our method of defining TATA versus paused genes among our late-expressed effector genes. By using CAGE data to re-annotate the TSS of our genes, we observed striking enrichments of different promoter elements in our groups. We then compared our late TATA and paused genes to housekeeping genes and developmentally paused genes as requested. As we had assumed but not explicitly discussed, this showed that the promoter type of late paused genes is essentially identical to that of developmentally paused genes. We have now made this clear in the revised version. In addition, we analyzed the differences between gene groups with regard to gene length, promoter shape and several expression variability measurements, which confirmed our previous conclusion that the TATA gene group has the most expression variability. Together with the remaining improvements described in our point-by-point response to the reviewers' comments, we feel that the revised manuscript is now strengthened in its conclusions and clarity.

Reviewer #1:

In the manuscript "TATA and paused promoters active in differentiated tissues have distinct expression characteristics", Ramalingam et al characterize expression properties of different promoter types of effector genes in late *Drosophila* embryo using scRNA-seq combined with profiling of polymerase pausing, chromatin accessibility and nucleosome occupancy. The study is important because it shows different modes of regulation of effector genes (genes responsible for the structure and function of differentiated tissues), which have been previously mostly linked to TATA-box promoters. In particular, authors propose that "highly paused promoters are more optimal for achieving low expression variability when genes are active, while TATA promoters are more optimal for achieving very low background expression when the genes are inactive". The manuscript is well-written, and results are supported by data. Yet, in parts, it is was not clear what the exact analyses where that the authors performed. See specific comments below. The main novelty of this work is the description, characterization and function of TATA-box and PolIII-pausing promoters in tissue-specific effector genes, thus excluding developmental and house-keeping genes. However, since much is known about the latter, especially in the context of TATA and PolIII, the study would benefit from a broader perspective by including these types of promoters as separate classes in their computational analysis where possible, to put their findings into a broader context. See more specific comments below:

Thanks for appreciating our study and for the suggestions.

Major comments:

1. One of the main conclusions of the study is that there are two fundamentally different modes of regulation within effector genes, as reflected by differences in polymerase pausing, nucleosome occupancy and expression noise of these genes. Also, the authors propose in the Discussion that "the two promoter types play different roles in evolution". Given these conclusions, the study would benefit from more in-depth characterization of functional differences between these different types of effector genes.

We agree that we had kept our definition and analysis of effector genes relatively short and have now expanded on this. We now better describe our analysis, added developmental paused genes, and

housekeeping genes as control groups and performed additional ones, e.g. the expression variability across *Drosophila* population isolates (based on Sigalova et al., 2020, MSB), as well as promoter shape (Fig EV1,2,3), all of which are consistent with our conclusions and those of other studies. One of the main takeaways that we had not emphasized before is that paused effector genes have essentially the same promoter type as paused developmental genes.

2. It was not clear what set of genes the authors actually take into account for their analyses - this reviewer was not familiar with the term 'effector genes'. They should provide their definition of "effector genes" and they should give numbers of how many genes these are the first time they talk about it (i.e. p2).

We define effector genes in the introduction as "genes responsible for the structure and function of differentiated tissues". In results, we then define them as genes that are specifically induced late in embryogenesis (to distinguish them from housekeeping and developmental genes, which are expressed earlier) and show that the functional GO categories of these 1,527 genes are consistent with their definition. We also include the number of genes analyzed in each promoter group in the main text and provide detailed Supplementary spreadsheets where interested readers can obtain more details on the exact nature of these genes (Dataset EV2, EV6).

3. More generally, the authors should consider including data for housekeeping and developmental genes (e.g. as background or comparison to effector genes) to put their findings into a broader context. This could make the results for effector genes stronger and also highlight novelty of the study. If not, at least it has to be more clearly stated, that only a subset of genes is analyzed (and number have to be given).

We are grateful for this suggestion and have now compared the late-induced TATA and paused genes to developmental paused genes and housekeeping genes in our analysis by various metrics (Fig EV1,2,3). This showed that highly paused effector genes and developmental genes have essentially the same promoter type. We therefore conclude that effector genes, as defined by their tissue-specific function, are not restricted to TATA genes, but also use the paused promoter, which is already used during the patterning stages of development.

Some interesting questions along these lines are:

a. Strong pausing was previously linked to developmental genes, while effector genes are mostly associated with TATA-box. Is there a difference in regulation / promoter architecture of highly paused effector vs developmental genes?

With the help of CAGE data, we have now analyzed the promoter type of paused effector genes and paused developmental genes in more detail. This showed that they are indistinguishable with regard to core promoter motifs, pausing index, gene length, and promoter shape (EV1). The functions of paused effector genes are however consistent with them being effector genes, which means that effector genes use both TATA promoters and highly paused promoters.

b. GAF (GAGA factor) has been previously linked to PolII pausing in *Drosophila* (Tsai et al 2016, PMID: 27468311). Here, the authors found the GAGA motif moderately enriched in all gene groups. What was the background for this enrichment? Does this imply that effector genes don't depend on GAF for pausing and there is an additional pausing factor for effector genes? Would the same enrichments be observed if GAF ChIP-seq was used in addition to GAGA-motif (since there is at least one other TF CLAMP known to bind GAGA-motif in promoters)? Can the authors distinguish "developmental pausing" from "effector pausing" by the GATA motif? Here a comparison with developmental genes would be helpful.

Thank you for pointing this out. The GAGA motif that we had used earlier in our enrichment analysis was too degenerate (literally GAGA). Using the stricter GAGAG motif, as we have done before in previous papers, now revealed that only the paused promoters were enriched for this motif (Fig 2C).

c. Li & Gilmour 2013 (PMID 23708796) propose two pausing mechanisms for M1BP-bound genes (mostly, housekeeping genes, transient pausing) and GAF-bound genes (mostly, developmental genes, strong pausing). Also, authors report similar results regarding pausing, nucleosome occupancy and expression variability for highly paused vs. TATA-box genes. How do gene groups identified here agree with those in (Li&Gilmour 2013)? The authors should cite this paper.

Our results agree with those of Li & Gilmour since the developmental genes and effector genes are not enriched for the Ohler1 motif, to which M1BP binds. As expected, our newly added housekeeping gene group is however enriched for Ohler1 motifs. We also now cite this paper for supporting the variability of TATA genes.

4. Can the authors exclude the that the pausing group of effector genes simply arose from an incomplete removal of developmental genes? (Sup Fig 3B shows Development as GO term in paused genes)

As we now discuss and better describe in the revised manuscript, paused effector genes and paused developmental genes share the same promoter type. However, a good fraction of paused effector genes clearly have functions that are very similar to the TATA effector genes. Thus, while some paused effector genes could be classified as developmental genes by different methods, there is strong evidence that effector genes, defined by their function in differentiated tissues, also use paused promoters. This means that paused promoters are used throughout embryogenesis, including in differentiated tissues.

5. Figure 3 presents some of the key results, and the authors should show more convincing examples in panel D such that the genes in the two classes have a similar mean expression level (the same way the genes were grouped by mean expression in panel A). In the current examples it is difficult to judge whether the background in the TATA genes is really different from the background in the pausing genes. Maybe the authors can pick 3 genes from three bins of expression level (low, medium high) to show this is true for all expression levels.

We have now included additional examples in the supplemental figures (Fig EV2F).

In addition the authors should visualize some aggregate information about background expression levels across genes (paused vs tata-box). The percentage of cells with expression shows only noise not really background expression levels.

We have now also quantified the background expression as mean transcript levels, which confirms our conclusions (Fig EV2E).

6. The methods description and/or figure legends miss some details regarding the analysis, data sources, sample sizes/significance for some group comparisons, which should be clarified in the main text and / or methods section. In particular:

- How are housekeeping and developmental genes defined (sources)? The statement: "By eliminating developmental genes and ubiquitously-expressed housekeeping genes from all late-expressed genes" has to be accompanied by some reference / or the actual list of genes that were removed.

We completely agree and have now rewritten the entire section to better define effector genes, how they were identified, and why we think they represent effector genes. We have also added control groups for housekeeping genes, and developmental genes. It now says in the results "To distinguish effector genes from housekeeping genes and developmental genes, we defined effector genes by their late upregulation during embryogenesis ($> 5x$, $p < 0.05$ from 2-4 h to 14-17 h), which yielded 1,527 genes (Dataset EV1, EV2, Methods). As control groups, we also defined ubiquitously-expressed housekeeping genes (647 genes), as well as developmental genes that are highly paused throughout embryogenesis (772 genes)(Dataset EV1) as defined previously (Gaertner et al. 2012)."

- Which genes are used as background for GO enrichments?

All annotated promoters in the genome were used as background.

- For the PolII tissue-specificity statement: "The Pol II occupancy was however not always tissue-specific. Some genes showed high Pol II occupancy at the promoter in many or all tissues, despite being expressed in a very tissue-restricted fashion." How many genes are we talking about in each class? Please cite the numbers in the text. In general, it would be appreciated to cite more numbers in the text.

We now mention the number of genes in the different groups both in the figure and in the main text.

- Fig 2c - the grouping is misleading, it seems the promoters are grouped by #of tissues with PolII enrichments, but the last column indicates this exactly corresponds to the TATA / pausing etc classification. In the text they mention it is enrichment, so they should show it as enrichment, rather than just colors. Also, what exactly is then the colors in Fig 2D/E corresponding to? - the actual pausing/ TATA classes, or the #tissues with PolII enrichment? Is it only one promoter group enriched for each of the 7 classes? If not, maybe some more quantitative visualisation could be helpful (e.g. groups in columns showing enrichment in rows - same as for promoter motifs).

We classified the promoter types based on the Pol II penetrance and then named these classes based on the enriched promoter motifs (without changing their contents). We have now added sequence maps with consensus motifs to show that this promoter classification is indeed convincing and not merely just a statistical enrichment (Fig 2D). We have also rewritten this section to improve clarity.

- Fig2d - it is not clear what expression data is shown. Please clarify in the legend.

This analysis is based on RNA-seq data from whole embryos aged 14-17 h. We have now clarified this in the figure legend (now found as a panel of EV1E to make space for the promoter sequence analysis).

- Fig2e - data shown on the x and y-axes seem to be redundant since it's Pausing index on y-axis and groups based on pausing on the x-axis. Might be more informative to show the 7 groups from 2c (based on the number of tissues with Pol II enrichment) on the x-axis.

As explained above, these are the groups defined by Pol II penetrance (x-axis is Pol II penetrance, y is pausing index) and we have clarified this in the figure legend (now part of EV1D).

- What are number of genes and significance of mean difference in panel 3a?

As mentioned below, we corrected the CV and it shows significantly higher variation for the TATA genes compared to the paused genes (Fig EV2A,B). We have also added p values (Fig EV2B).

- Gene expression noise (referring to Fig3a) - this metric is usually calculated by adjusting coefficient of variation for expression level since there is strong negative relationship between CV and expression level (this is seen for highly paused genes). While probably results in panel 3a won't be affected, this adjustment would allow for more accurate comparisons of noise levels among gene groups (e.g. for estimating statistical significance of differences in noise between paused and TATA genes).

We have now corrected the coefficient of variation using loess regression. Corrected CV also shows significantly higher variation for the TATA genes compared to the paused genes (Fig EV2A,B).

Minor points:

- Is Figure 1B really the best visualization for the single cell data? It is somewhat difficult to spot the data within all the lines and the schematics.

That is true, but the alternative is to give each cluster a number and then place these numbers on the *Drosophila* tissues. This makes it even harder to find the tissues. We had visually played with the figure for a while and found this to be the best solution.

- Fig3b,c - what was the threshold on expression level to define expressing cells?

Cells with detectable transcripts (>1 read) are considered as cells with expression, which is now mentioned in the main text and described in the methods. We have chosen this threshold after analyzing the effect with different thresholds. In the expressing tissue, using higher cutoffs gives similar results. In the non-expressing tissues, using higher cutoffs leads to noisier results, since the background expression is generally low, which is why we chose the low cutoff of 1 read.

- Fig4a - are those genes expressed at both time-points (2-4 and 14-17h)? Is the difference in accessibility significant for dual TATA and paused genes? Please clarify in the text.

All effector genes were selected for being expressed in late embryos but not early embryos, which we now mention when we introduce the identification of effector genes. The difference in accessibility between dual TATA and paused genes is significant, which is not shown. Generally, we only calculated the significance between the highly paused and TATA gene groups.

- p5: "We found that the TATA genes had indeed significantly fewer annotated expression patterns compared to..." - is the end of the sentence missing? Shall this refer to panel 3F?

Thank you for pointing out this formatting issue, which we have now fixed. It now says "We found that the TATA genes had significantly fewer annotated expression patterns compared to highly paused genes".

- p2: header should specify that this is effector genes

Thank you. The header now says "Effector genes have different Pol II occupancy patterns across tissues".

Reviewer #2:

The manuscript by Ramalingam et al presents evidence that TATA-dependent tissue-specific promoters in *Drosophila* have distinct properties from TATA-less tissue-dependent promoters with respect to PolII

pausing, cell-to-cell variability of expression levels, and background "leaky" expression in the cells and tissues in which the gene is (mostly) inactive. They used single cell RNA-seq to determine tissue specificity, variability and background levels of expression of individual genes, and tissue-specific PolII ChIP-seq to determine PolII pausing behaviour at each.

This is a very nice body of work that shows exactly what it claims it is showing, using elegant, cutting-edge methodology, and it should be of great interest to others in the field. What it would benefit from is better integration with what is already known about the differences between TATA-dependent and TATA-independent promoters, both in *Drosophila* and in other organisms:

Thank you for appreciating the novelty of our study and for the constructive suggestions.

- The authors removed "developmental" promoters to compare TATA-dependent and TATA-independent "late" tissue-specific promoters, and found differences. I believe that there is a flaw here. DPE- and pause-button-dependent promoters differ from TATA-dependent promoters primarily with regard to their dependence on long-range enhancers. In that sense, the TATA-less, DPE- or pause-button containing tissue-specific promoters will probably be a part of the continuum of developmental promoters - just those whose associated enhancers are active only relatively late in development and differentiation, and in a small number of tissues. Other than that, I would expect their properties to be similar to, or even indistinguishable from, developmental promoters active early in development or in multiple tissues. For that reason, it would be essential to extend the analysis to the removed developmental promoters to examine this relationship.

Yes, exactly and thanks for pointing this out. We had assumed and now show, that the promoters of developmental genes are essentially the same as those of highly paused effector genes. This shows that classifying genes by developmental stage or function is insufficient to separate promoter types and that promoters with paused Pol II continue to be used beyond the developmental stages. We now compare the effector genes to a group of developmental genes (as defined by continuous Pol II pausing throughout development by Gaertner et al., 2013) and explicitly say that their promoters are indistinguishable by several criteria. We also mention in the discussion the presumed difference between TATA and paused genes with regard to long-range enhancers.

- Indeed, even though the authors claim that there is no clear difference in GO terms between TATA and highly paused genes, it is no surprise to me at all that "developmental process" and "cell communication", both terms known to be associated with multicellular processes under long-term enhancer regulation, show up with the highly-paused genes. If more highly paused genes are included, the overrepresentation of the two terms will only become stronger.

We agree and now discuss our GO analysis in more detail. The GO analysis does not preclude the possibility that paused effector genes have functions similar to traditional developmental genes. It does however suggest that a good fraction of paused effector genes have tissue-specific functions consistent with them being effector genes. This led us to reject the hypothesis that only TATA genes are effector genes.

- The difference with respect to the dependence on long-range enhancers will have several consequences that the author observe, and are expected: in a secondarily compacted genome such as that of *Drosophila*, the genes with TATA-dependent promoters will have shorter introns, and more likely to occur as tandem duplicates. On the other hand, DPE-dependent promoters, as essentially developmental promoters often controlled by multiple enhancers, will often have longer introns and intergenic spaces

needed to accommodate those enhancers, and will be more dosage-sensitive, both of which would make tandem duplication disruptive with respect to enhancer-promoter interactions.

We completely agree and have incorporated some of these ideas in our discussion. It now says “The large number of enhancers makes it more likely that gene duplications disrupt enhancer-promoter interactions and that promoter mutations have detrimental pleiotropic effects..”

- Some previous research suggests that, unlike developmental promoters, most TATA-dependent promoters in vertebrates are actually regulated by proximal cis-regulatory modules just upstream of the core promoters (e.g. Roeder HG et al NAR 2009 showed that the difference between TATA-dependent and TATA-independent liver- and muscle-specific genes is that the non-CpG, TATA-dependent ones are the only ones that contain an enrichment of tissue-specific motif at the proximal promoters; Soler E et al Genes Dev 2010 show a rather dramatic difference in the relative position of GATA1 binding between the two types of erythroid-constrained promoters in mouse). This would suggest the difference in mechanism between TATA-dependent and developmental, enhancer-driven promoters: the former are fully inactive until a context-specific transcription factor binds to them; the latter are poised to respond to enhancer activation, which may be accomplished by starting the transcription and pausing it until enhancer interaction releases the pause.

We completely agree. We have now extended the discussion and cited these papers. It now says “These genes may correspond to the simple “gene batteries” described by Eric Davidson and may be regulated predominantly by promoter-proximal regulatory elements (Erwin and Davidson 2009; Roeder et al. 2009; Soler et al. 2010). In contrast, promoters with paused Pol II tend to be found in relatively long genes with extensive cis-regulatory regions.”

- The chromatin accessibility of TATA promoters will be influenced by their two properties that are already well known: 1) the nucleosome positions at TATA promoters are not as precise as at most non-TATA promoters, because it is the TATA box position that directs the TSS selection to about 30 bp downstream of it, regardless of the stable position of the nucleosomes at each particular promoter (see e.g. Rach E et al PLOS Genet 2011, and a detailed analysis in Dreos R et al PLOS Comp Biol 2016). It might be that the positions are relatively stable at individual promoters, but since they are not at a constant distance from the TSS, there is no reference they line up with in heatmaps or metaplots. Alternatively, since TATA-dependent promoters act in large "bursts" with periods of inactivity between the bursts, the nucleosome arrangement will be a time average between active and inactive states.

These are again very thoughtful comments. We now mention the presence of fuzzy nucleosomes and longer transcriptional bursts at TATA promoters. It now says, “TATA promoters have fuzzy promoter nucleosomes (Tirosh and Barkai 2008; Gaertner et al. 2012)” and “Transcription of many genes is a discontinuous process, resulting in bursts of transcripts (Coulon et al. 2013; Raj et al. 2006). These bursts of transcription are larger at TATA promoters (Larsson et al. 2019; Hornung et al. 2012; Tantale et al. 2016).” We agree however with the reviewer that the exact nature of the fuzzy nucleosomes at TATA promoters is not clear: it could be due to heterogeneity of nucleosome positioning between different TATA promoters or due to heterogeneity between promoters in the active and inactive state. The first should result in averaging out of the nucleosome signal, while the second should result in lower nucleosome occupancy. We therefore favor the first explanation, but both likely occur, and we did not want to unnecessarily speculate in the manuscript.

- More generally, what is missing is the analysis of additional promoter features that are known to be associated with different promoter types. It is unclear how the authors defined transcription start sites, but

defining them by high-quality CAGE data (from modENCODE or from Eileen Furlong's lab) is more likely to reveal precise spatial dependencies between transcription start sites and motifs such as TATA-box and DPE, as well as MNase signal and any nucleosome positioning signals in the sequence, if present. Also, the authors do not mention if the analysed promoters are predominantly "peaked" (single TSS position) or "dispersed" (multiple TSS positions within the same promoter region). TATA- and DPE-dependent promoters are generally peaked with constrained spacing between the TSS and the motif (e.g. Dreos R PLOS Comp Biol 2016).

This was an excellent suggestion. We have now used CAGE data to better define the TSS and this has allowed us to better define the consensus motifs present in our promoters (see Fig 2 and EV1). We have also classified the promoters as broad versus narrow and show that effector genes are predominantly narrow as expected.

-Currently, there is not enough evidence to support the author's claim that accessibility does not change between expressing and non-expressing tissues as there is no direct comparisons but rather very global trends. It would be interesting to directly compare accessibility in expressing and non-expressing tissues for the same genes, as heatmaps with genes separated into their tissue of maximum expression.

We performed additional analysis as suggested and observed a trend between tissue-specific expression and tissue-specific accessibility. However, we felt that the correlation was not strong enough to make strong claims and thus have not added these results to the paper.

Minor:

- It would be helpful if the authors could clearly define Pol II penetrance in the main text, how it is calculated and how it differs from the pausing index it is compared to.

We have defined Pol II penetrance as the number of tissues (from 0 to 6 tissues) in which Pol II is detected around the transcription start site (200bp downstream of the TSS to the TSS) to be above background (>2 fold signal over input). This resulted in four groups: TATA-enriched (527 genes in 0 tissues), dual TATA (415 genes in 1-2 tissues), paused (222 genes in 3-4 tissues) and highly paused (362 genes in 5-6 tissues). There is good correspondence between our Pol II penetrance groups and pausing index (EV1D). The pausing index by itself is however a noisy measurement (since quotient is a very small number), which is why we did not use it as a primary means of classifying promoters.

- The motif identification allowing zero mismatches is quite stringent. Would the enrichments apply if lower thresholds were applied, for example 90% match to the PWM?

This is indeed a stringent threshold. It underestimates the number of promoter elements present, but accurately captures enrichments, which is the purpose of the plot. To better capture the promoter elements from all genes in the group, we now show their raw sequence as a heat map and derive consensus motifs for all groups de novo (thanks to the better TSSs defined by CAGE). This shows very clearly the high fraction of promoters with TATA-like elements or pausing elements, respectively.

- In Figure 3 it is not immediately clear what the threshold for expression is for a cell to be classified as expressing.

Cells with detectable transcripts (at least one read) are considered as cells with expression. This information is now found both in the methods and the figure legend. We have also analyzed the effect of

different thresholds on our frequency of expressing cells measurements. In the expressing tissue, using higher cutoffs gives similar results. In the non-expressing tissues, using higher cutoffs leads to noisier results, since the background expression is generally low.

- In figure 4D it would be better if the heatmaps were ordered by accessibility (highly accessible regions at the top, low accessibility regions at the bottom). It would be easier to assess whether higher accessibility is a global feature of these promoters or if few regions with very high accessibility are skewing the trend.

We have now changed the figure to the sorted version. We have also calculated the statistical significance of the average signal across tissues between groups (Wilcoxon two-sided test).

- In Figure S4, can the authors comment on why we do not see highly paused promoters reflected here, with consistent Pol II occupancy in all tissues?

This comparison between Pol II occupancy and RNA expression was done to examine the tissue-specific purity of the Pol II sample. Therefore, we only included genes where Pol II occupancy was found in a single tissue. Paused promoters with Pol II in many tissues are not expected to have a high correlation with expression and thus confound the analysis. Nevertheless, even if we use all effector genes, the tissue with the highest Pol II occupancy is in most cases the tissue with the highest RNA levels.

- In supplementary excel 3, it is not clear what is denoted in new_start and new_end columns.

Thank you for catching this. We have now added legends and removed unnecessary columns (now Appendix Table 2).

Reviewer #3:

This paper describes the relationship between promoter types and expression features in the late *Drosophila* embryo. First, single cell RNA-seq (scRNA-seq; with cells derived from whole embryos) was used to obtain gene expression data in various tissues (Fig. 1). Then, by using tissue-specific ChIP-seq with six different tissues, the Pol II occupancy in different tissues was determined (Fig. 2). Genes with high Pol II 'penetrance' (i.e., widespread Pol II occupancy in different tissues) were found to be noisier when inactive and less variable when active, relative to genes with low Pol II penetrance (Fig. 3). Chromatin accessibility also correlates with Pol II penetrance (Fig. 4).

This manuscript is clearly written, and the experiments were well designed. If the Major comment is addressed appropriately, publication would be recommended.

Thank you for appreciating our study and for the constructive suggestions.

Major comment:

1. In Fig. 2C, the authors classify genes based on the occurrence of enriched Pol II (from the +1 TSS to +200) in different tissues. Accordingly, genes in which Pol II is enriched in 5/6 or 6/6 tissues are called "highly paused". In contrast, genes in which Pol II is not enriched (in 0/6 tissues) are called "TATA". In the "TATA" genes, the TATA-box appears to be enriched by about 4 fold; however, it is likely that many of the "TATA" genes lack a TATA box. Therefore, unless all of the "TATA" genes contain a TATA box (and all of the "Highly Paused" genes lack a TATA box), the "TATA" gene category should be re-named because the

name would be inaccurate and misleading. One possible alternate name would be "Non-paused", as used in Fig. 2B. This name would match the basis upon which the genes were categorized. It could be stated that the "Non-paused" genes are enriched for TATA boxes.

This is a good point. As we show in the revised manuscript as sequence color maps and consensus motifs, the majority of genes in the TATA group indeed have a TATA-like element, while the majority in the highly paused group show a downstream promoter element. We therefore feel that our grouping by "Pol II penetrance" is a very good way of identifying promoter groups, while using the presence of motif consensus binding motifs is less reliable. However, we agree that calling the group "TATA-enriched" may be less confusing and have now renamed this group.

Other comments:

2. Please specify exactly how the scRNA-seq data were linked to the tissue-specific Pol II occupancy. It is not obvious that there is a good match between the tissues in the two different experiments. No correlation value is shown in Fig. S4, which could help establish the link. In addition, the data presented in Fig. S4 are restricted to genes with Pol II occupancy in a single tissue.

The comparison was indeed done with genes that have Pol II occupancy in a single tissue since paused promoters are not expected to have a high correlation with their expression. But with or without this restriction, the correlation coefficient is only medium and thus not very informative, presumably because the low values are noisy. For this reason, we now include the percentage of genes for which the tissue with the highest Pol II matches the tissue with the highest expression (~77% of genes with Pol II occupancy in a single tissue).

3. In Fig. 3B-C, it would be useful to show the distribution of normalized RNA levels in addition to the percent of expressing cells. For example, a 2D plot of Normalized RNA levels vs. % Cells with Expression might be informative. "Cells with expression" should also be clearly defined [there is a similar issue in Fig. 2 for "(non-)expressing tissues"].

We have plotted (RNA levels vs. % Cells with expression) in Fig EV2. Cells with detectable transcripts (>1 read) are considered as cells with expression, which we have now clarified in the legends and the methods. We have however also analyzed the effect of different thresholds on our frequency of expressing cells. In the expressing tissue, using higher cutoffs gives similar results. In the non-expressing tissues, using higher cutoffs leads to noisier results, since the background expression is generally low.

4. In the legend of Fig. 2, it is stated that the differences in paused Pol II occupancy are related to sample preparation and transcription. Please explain.

Thank you for asking this question. While the highly paused genes have paused Pol II in all tissues, the levels vary. A clear trend is that paused Pol II is highest in the tissue with the highest expression (in ~55% of highly paused genes). But we also found that there are systematic differences between samples, thus some tissues have generally higher enrichments than others. We speculate that this is because tissues that are outside (e.g. epidermis) are easier to crosslink. We have now extended our explanation in the legend: "Paused Pol II is generally highest in the tissue with the highest expression. We also found systematic differences between samples, thus some tissues have generally higher enrichments than others, presumably because they are easier to crosslink."

14th Dec 2020

Manuscript Number: MSB-20-9866R

Title: TATA and paused promoters active in differentiated tissues have distinct expression characteristics

Author: Vivekanandan Ramalingam

Malini Natarajan

Jeff Johnston

Julia Zeitlinger

Thank you for sending us your revised manuscript. We have now heard back from two of the three reviewers who were asked to evaluate your study. Unfortunately, after a series of reminders we did not manage to obtain a report from Reviewer #2. In the interest of time, and since the other two reviewers' recommendations are quite similar, I prefer to make a decision now rather than further delaying the process. As you will see the reviewers are satisfied with the modifications made and think that the study is now suitable for publication.

Before we can formally accept your manuscript, we would ask you to address a few remaining editorial issues listed below:

1. Please provide up to 5 keywords and incorporate them in the main text.
2. A Conflict of Interest statement should be provided in the main text.
3. Appendix Table 1 & 2: Please add an 'S' to Appendix Table (Appendix Table S1).
4. Reference format: list all 10 co-authors of a paper before to add et al. in the reference list. More information can be found here (<https://www.embopress.org/page/journal/17444292/authorguide>).
5. Appendix: add a Table of Content on the 1st page. Please note that this file will not be typeset nor proofread.
6. Funding: Funding information was entered in the online submission system, but absent from the manuscript. Please add it to the manuscript.
7. EV datasets : please uploaded them individually with each individual legend inserted into each file in a separate sheet.
8. Our data editor has made some comments to your manuscript (please see attached). Please address these issues and keep the track mode on.
9. Synopsis image: the text becomes somewhat blurry when the image is adjusted to the required size (550 px width). Please provide a new image (JPEG format, 550 px width and ~400 px high) with higher resolution text.

10. I have made some modifications to the synopsis text. Please let me know if it is fine like this or if you would like to introduce further modifications.

This study characterizes expression properties of different promoter types of effector genes in late-stage *Drosophila* embryos, using scRNA-seq combined with profiles of Pol II occupancy, chromatin accessibility and nucleosome occupancy.

- Paused promoters show high levels of Pol II pausing throughout the embryo and have low expression variability but high background expression.
- TATA promoters show tissue-specific Pol II recruitment and low chromatin accessibility, and have low background expression but high expression variability.
- Precise gene expression requires both low background expression and low expression variability.

When you resubmit your manuscript, please download our CHECKLIST (<https://bit.ly/EMBOPressAuthorChecklist>) and include the completed form in your submission. *Please note* that the Author Checklist will be published alongside the paper as part of the transparent process (<https://www.embopress.org/page/journal/17444292/authorguide#transparentprocess>)

Click on the link below to submit your revised paper.

Sincerely,
Jingyi

Jingyi Hou
Editor
Molecular Systems Biology

If you do choose to resubmit, please click on the link below to submit the revision online before 13th Jan 2021.

Link Not Available

IMPORTANT: When you send your revision, we will require the following items:

1. the manuscript text in LaTeX, RTF or MS Word format
2. a letter with a detailed description of the changes made in response to the referees. Please specify clearly the exact places in the text (pages and paragraphs) where each change has been made in response to each specific comment given
3. three to four 'bullet points' highlighting the main findings of your study

4. a short 'blurb' text summarizing in two sentences the study (max. 250 characters)
5. a 'thumbnail image' (550px width and max 400px height, Illustrator, PowerPoint or jpeg format), which can be used as 'visual title' for the synopsis section of your paper.
6. Please include an author contributions statement after the Acknowledgements section (see <https://www.embopress.org/page/journal/17444292/authorguide#manuscriptpreparation>)
7. Please complete the CHECKLIST available at (<https://bit.ly/EMBOPressAuthorChecklist>). Please note that the Author Checklist will be published alongside the paper as part of the transparent process (<https://www.embopress.org/page/journal/17444292/authorguide#transparentprocess>).
8. Please note that corresponding authors are required to supply an ORCID ID for their name upon submission of a revised manuscript (EMBO Press signed a joint statement to encourage ORCID adoption) (<https://www.embopress.org/page/journal/17444292/authorguide#editorialprocess>).

Currently, our records indicate that the ORCID for your account is 0000-0002-5172-3335.

Link Not Available

The system will prompt you to fill in your funding and payment information. This will allow Wiley to send you a quote for the article processing charge (APC) in case of acceptance. This quote takes into account any reduction or fee waivers that you may be eligible for. Authors do not need to pay any fees before their manuscript is accepted and transferred to the publisher.

*** PLEASE NOTE *** As part of the EMBO Press transparent editorial process initiative (see our Editorial at <https://dx.doi.org/10.1038/msb.2010.72> , Molecular Systems Biology will publish online a Review Process File to accompany accepted manuscripts. When preparing your letter of response, please be aware that in the event of acceptance, your cover letter/point-by-point document will be included as part of this File, which will be available to the scientific community. More information about this initiative is available in our Instructions to Authors. If you have any questions about this initiative, please contact the editorial office (msb@embo.org).

Reviewer #1:

The authors have addressed all our comments. Congratulations on this very-well presented and stimulating study!

Reviewer #3:

The authors have addressed all of my concerns. Publication in MSB is recommended.

The authors performed the requested changes.

7th Jan 2021

Manuscript number: MSB-20-9866RR

Title: TATA and paused promoters active in differentiated tissues have distinct expression characteristics

Thank you again for sending us your revised manuscript. We are now satisfied with the modifications made and I am pleased to inform you that your paper has been accepted for publication.

*** PLEASE NOTE *** As part of the EMBO Publications transparent editorial process initiative (see our Editorial at <https://dx.doi.org/10.1038/msb.2010.72>), Molecular Systems Biology publishes online a Review Process File with each accepted manuscripts. This file will be published in conjunction with your paper and will include the anonymous referee reports, your point- by-point response and all pertinent correspondence relating to the manuscript. If you do NOT want this File to be published, please inform the editorial office at msb@embo.org within 14 days upon receipt of the present letter.

Should you be planning a Press Release on your article, please get in contact with msb@wiley.com as early as possible, in order to coordinate publication and release dates.

LICENSE AND PAYMENT:

All articles published in Molecular Systems Biology are fully open access: immediately and freely available to read, download and share.

Molecular Systems Biology charges an article processing charge (APC) to cover the publication costs. You, as the corresponding author for this manuscript, should have already received a quote with the article processing fee separately.

Please let us know in case this quote has not been received.

Once your article is at Wiley for editorial production you will receive an email from Wiley's Author Services system, which will ask you to log in and will present you with the publication license form for completion. Within the same system the publication fee can be paid by credit card, an invoice or pro forma can be requested.

Payment of the publication charge and the signed Open Access Agreement form must be received before the article can be published online.

Molecular Systems Biology articles are published under the Creative Commons licence CC BY, which facilitates the sharing of scientific information by reducing legal barriers, while mandating attribution of the source in accordance to standard scholarly practice.

Proofs will be forwarded to you within the next 2-3 weeks.

Thank you very much for submitting your work to Molecular Systems Biology.

Sincerely,

Jingyi Hou
Editor
Molecular Systems Biology

Corresponding Author Name: Julia Zeitlinger

Manuscript Number: MSB-20-9866